# CoRA: Boosting Time Series Foundation Models for Multivariate Forecasting through Correlation-aware Adapter

**Hanyin Cheng**[1], **Xingjian Wu**[1], **Yang Shu**[1], **Zhongwen Rao**[2], **Lujia Pan**[2]
**Bin Yang**[1], **Chenjuan Guo**[1✉]
[1]East China Normal University
[2]Huawei Noah's Ark Lab
```
{hycheng,xjwu}@stu.ecnu.edu.cn,
{raozhongwen,panlujia}@huawei.com,
{yshu,byang,cjguo}@dase.ecnu.edu.cn,
```

## Abstract

Most existing Time Series Foundation Models (TSFMs) use channel independent modeling and focus on capturing and generalizing temporal dependencies, while neglecting the correlations among channels or overlooking the different aspects of correlations. However, these correlations play a vital role in Multivariate time series forecasting. To address this, we propose a **CoR**relation-aware **A**dapter (**CoRA**), a lightweight plug-and-play method that requires only fine-tuning with TSFMs and is able to capture different types of correlations, so as to improve forecast performance. Specifically, to reduce complexity, we innovatively decompose the correlation matrix into low-rank Time-Varying and Time-Invariant components. For the Time-Varying component, we further design learnable polynomials to learn dynamic correlations by capturing trends or periodic patterns. To learn positive and negative correlations that appear only among some channels, we introduce a novel dual contrastive learning method that identifies correlations through projection layers, regulated by a Heterogeneous-Partial contrastive loss during training, without introducing additional complexity in the inference stage. Extensive experiments on 10 real-world datasets demonstrate that CoRA can improve TSFMs in multivariate forecasting performance.

## 1 Introduction

Time Series Foundation Models (TSFMs) that show strong generalization are proposed recently. Through pre-training on large-scale time series data (Wang et al., 2025d; Goswami et al., 2024; Liu et al., 2024e; Wang et al., 2025e) or the use of large language models (Zhou et al., 2023; Liu et al., 2024d;b; Jin et al., 2024), these models maintain strong reasoning abilities when handling new or unseen data.

At the same time, multivariate time series forecasting, as a pivotal domain in data analysis, is widely applied in various industries (Qiu et al., 2025c; Wu et al., 2025c; Liu et al., 2026a; Qiu et al., 2025e; Liu et al., 2026b; Wang et al., 2024b; Qiu et al., 2025b). Properly modeling and utilizing correlations in multivariate time series can significantly improve the performance of forecasting models (Zhang & Yan, 2022; Liu et al., 2024c; Wu et al., 2020). Based on different interaction characteristics among channels, as shown in Figure 1a, correlation can be summarized into three aspects: *dynamic correlation (DCorr)* describes the variation of channel relationships over time (Zhao et al., 2023; Cirstea et al., 2021); *heterogeneous correlation (HCorr)* focuses on how channels interact with each other by considering positive and negative correlations (Huang et al., 2023); *partial correlation (PCorr)* emphasizes that correlation exists only among certain channels, and modeling interactions across all channels can easily introduce noise (Chen et al., 2024; Qiu et al., 2025d; Liu et al., 2024a). Considering more comprehensive correlations provides richer information for the models.

However, most existing TSFMs focus on capturing and generalising temporal dependencies and neglect relationships among channels (Goswami et al., 2024; Ansari et al., 2024; Liu et al., 2024e;b; Jin et al., 2024; Shi et al., 2025). Although some models like TTM (Ekambaram et al., 2024b), UniTS (Gao et al., 2024), and Moirai (Woo et al., 2024) employ different methods to model the correlations among channels, they do not comprehensively consider multiple aspects of the correlations. For example, TTM employs an MLP-based channel mixing approach, but the MLP weights remain unchanged across different time steps, thereby failing to model DCorr while indiscriminately modelling interactions among all channels, and thus failing to capture HCorr and PCorr explicitly.

While the attention mechanisms used in UniTS and Moirai assign different attention scores at different time points, they still interact all channels simultaneously without considering HCorr and PCorr, thus leading to suboptimal correlation modeling. Furthermore, due to the variations in correlations across different datasets, it is difficult to capture generalized correlations during the pre-training phase (Ekambaram et al., 2024b).

Thus, it motivates us to design a plugin that can be fine-tuned alongside TSFMs, which avoids issues caused by correlation differences across datasets during the pre-training phase. Meanwhile, it possesses

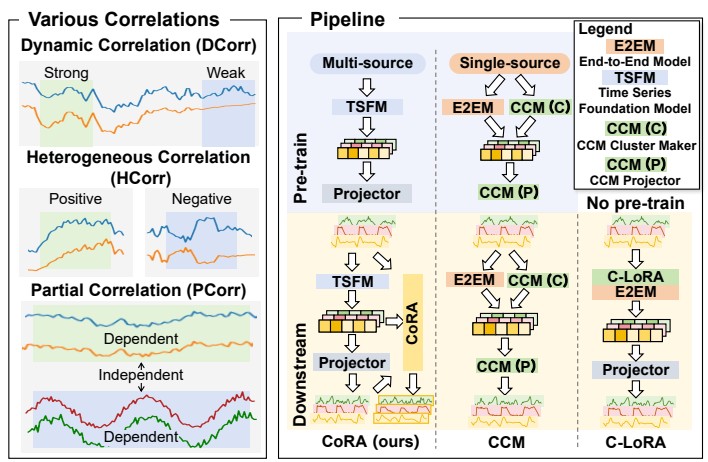

(a) Various Correlations  (b) Efficient Plugins for Learning Correlations

Figure 1: (a) Illustration of three different types of correlations, the formal definitions are provided in Appendix A. (b) Comparisons of different plugins for learning correlations.

the ability to depict various correlations while also incorporating a lightweight design. However, this faces a major challenge: **balancing the complete modeling of various correlations with the lightweight design.** It is intrinsically difficult to model all three correlations in a unified manner. Although some models could address DCorr (Zhao et al., 2023; Cirstea et al., 2021), HCorr (Huang et al., 2023) and PCorr (Qiu et al., 2025d; Liu et al., 2024a) individually, they struggle to effectively encompass various correlations simultaneously. Moreover, existing channel interaction methods often rely on MLPs (Ekambaram et al., 2023; 2024a), Transformers (Liu et al., 2024c; Jiang et al., 2023) and GNNs (Wu et al., 2020; Cai et al., 2024), etc., which have a time complexity of $\mathcal{O}(N^2)$, where $N$ denotes the number of channels. Some methods (Zhang & Yan, 2022; Chen et al., 2024; Nie et al., 2024) have made efforts in reducing the complexity. However, end-to-end models such as Crossformer (Zhang & Yan, 2022) require modifying or redesigning the entire model structure, and thus cannot be directly used as plugins for TSFMs. Existing plugins are primarily designed for end-to-end forecasting models. CCM (Chen et al., 2024) requires additional pre-training together with the end-to-end models before it can be plugged in. C-LoRA (Nie et al., 2024) is designed to be trained with an end-to-end backbone from scratch. Overall, there is a lack of an efficient plugin specifically designed for downstream fine-tuning of TSFMs. More importantly, considering various correlations in these methods would lead to a higher complexity.

To address this, we propose CoRA, a lightweight plug-and-play method that only requires training on a few samples with TSFMs during the fine-tuning phase. By considering various correlations, CoRA utilises internal representations and original predictions from TSFMs to generate an enhanced prediction, as shown in Figure 1b. To complete modeling the mentioned three types of correlation, we first propose the **Dynamic Correlation Estimation (DCE)** module which can learn dynamic correlation matrices. Then we design the **Heterogeneous-Partial Correlation Contrastive Learning (HPCL)** that uses the correlation matrices from DCE to learn HCorr and PCorr adaptively. Specifically, to achieve lightweight, we innovatively decompose the correlation matrices into two low-rank components: Time-Varying and Time-Invariant in DCE module. To better understand how DCorr evolves, we propose a learnable polynomial to capture trend or periodic patterns within the DCorr effectively. Afterwards, to better distinguish of HCorr, we propose channel-aware projec-

tions to map the representations into positive and negative correlation spaces. The projections are guided by the novel Heterogeneous-Partial Contrastive Loss during the training process, which enables adaptive learning of PCorr in the two HCorr spaces. As a result, we can capture the mentioned three types of correlations with linear complexity w.r.t. the number of channels during inference. Our contributions are summarized as follows:

- We design a universal, lightweight plugin that allows TSFMs to capture the mentioned three types of correlations without re-pre-training the TSFMs.
- We propose a lightweight Dynamic Correlation Estimation module that explicitly models the dynamic patterns of correlations in a lightweight manner.
- We propose a novel Heterogeneous-Partial Correlation Contrastive Learning, which learns HCorr and PCorr through projection layers regulated by dual contrastive loss.
- We conducted extensive experiments on 10 real-world datasets. The results show that CoRA effectively improves the performance of TSFMs in multivariate forecasting.

## 2 RELATED WORK

### 2.1 FOUNDATION MODELS FOR TIME SERIES FORECASTING

TSFMs for forecasting can be divided into two sections: **1) LLM-based Models:** These methods leverage the strong representational capacity and sequential modeling capability of LLMs to capture complex patterns for time series modeling. Among them, GPT4TS (Zhou et al., 2023) and CALF (Liu et al., 2025) selectively modify certain parameters of LLMs to enable the model to adapt to time series data. On the other hand, UniTime(Liu et al., 2024b), S$^2$IP-LLM (Pan et al., 2024), LLMMixer (Kowsher et al., 2025), and Time-LLM (Jin et al., 2024) focus on creating prompts to trigger time series knowledge within LLMs. **2) Time Series Pre-trained Models:** Pre-training on multi-domain time series equips these models with strong generalization capabilities. Among them, ROSE (Wang et al., 2024a) and Moment (Goswami et al., 2024) restore the features of time series data, enabling them to extract valuable information in an unsupervised manner. On the other hand, TimesFM (Das et al., 2024) and Timer (Liu et al., 2024e), using an autoregressive approach, employ next-token prediction to learn time series representations. Generally speaking, most TSFMs are based on channel-independent strategies, with only a few (Gao et al., 2024; Ekambaram et al., 2024b; Woo et al., 2024) modeling relatively simple inter-channel relationships. The effects of more complex correlations in TSFMs remain under-explored.

### 2.2 CORRELATION OF CHANNELS IN TIME SERIES FORECASTING

Channel correlation plays a crucial role in enhancing the predictions(Qiu et al., 2025a). They can be divided into specialized models and plugins from a paradigm perspective. **1) Correlation Models:** These models are typically based on foundational architectures such as MLP (Ekambaram et al., 2024a; 2023), GNN (Shang et al., 2021; Cai et al., 2024), and Transformer (Liu et al., 2024c; Zhang & Yan, 2022). For example, TSMixer(Ekambaram et al., 2023) and TTM(Ekambaram et al., 2024a) directly mix all channels using MLP. MTGNN(Wu et al., 2020) and Ada-MsHyper(Shang et al., 2024a) treat different channels as distinct nodes, performing message passing to facilitate channel interactions. Furthermore, iTransformer(Liu et al., 2024c) and Crossformer(Zhang & Yan, 2022) treat different channels as distinct tokens and utilize transformers to realize channel interaction. **2) Correlation Plugins:** Some plugins enhance the predictive capability of models by learning correlation (Cirstea et al., 2022; 2021). For example, LIFT (Zhao & Shen, 2024) leverages locally relationships to extract correlations. CCM (Chen et al., 2024) further performs clustering and creates dedicated prediction heads for each cluster. However, the methods above either lack comprehensive correlation modeling capabilities or possess substantial complexity.

## 3 PRELIMINARIES

**Time Series Forecasting.** Given a multivariate time series $\mathbf{X}_t = (\mathbf{x}_{t-L:t}^i)_{i=1}^N \in \mathbb{R}^{N \times L}$ with a look-back window $L$ and $N$ channels, the forecasting task aims to predict the future $F$ steps $\hat{\mathbf{Y}}_t = (\hat{\mathbf{x}}_{t:t+F}^i)_{i=1}^N \in \mathbb{R}^{N \times F}$, corresponding to the ground truth $\mathbf{Y}_t = (\mathbf{x}_{t:t+F}^i)_{i=1}^N$.

**Correlation-Aware Adapter for Foundation Models.** Consider a pre-trained Time Series Foundation Model (TSFM), denoted by $\mathcal{F}$. During the downstream fine-tuning phase, the model processes the input data $\mathbf{X}_t^{ft}$ to generate initial predictions, formulated as $\hat{\mathbf{Y}}_t^{ft} = \mathcal{F}(\mathbf{X}_t^{ft})$. Concurrently, the TSFM extracts the intermediate series representation, denoted as $\tilde{\mathbf{X}}_t^{ft}$. Given the downstream input $\mathbf{X}_t^{ft}$, its corresponding ground truth $\mathbf{Y}_t^{ft}$, the initial predictions $\hat{\mathbf{Y}}_t^{ft}$, and the latent representations $\tilde{\mathbf{X}}_t^{ft}$, a correlation-aware adapter is to fine-tune the base model $\mathcal{F}$ to yield an adapted model $\mathcal{F}^*$. Here, $\mathcal{F}^*$ represents the foundation model augmented with the proposed adapter. During the inference phase, predictions on unseen test data are generated via $\hat{\mathbf{Y}}_t^{test} = \mathcal{F}^*(\mathbf{X}_t^{test})$.

# 4 METHODOLOGY

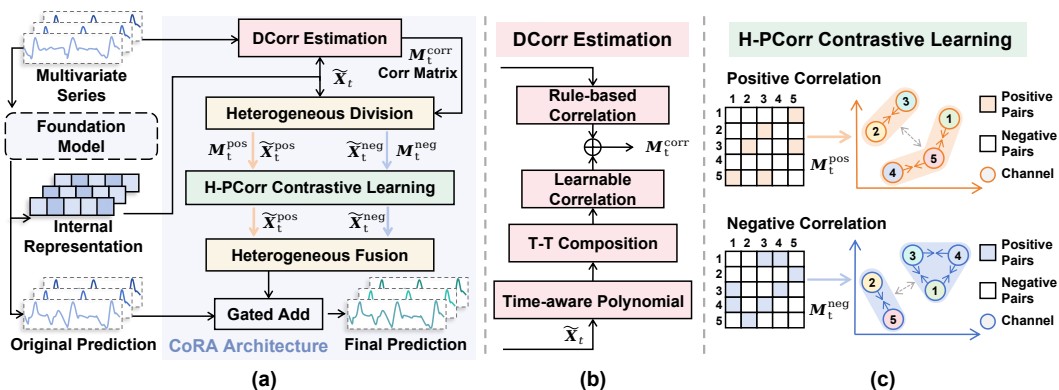

Figure 2: The framework of CoRA. (a) CoRA begins by learning DCorr in Dynamic Correlation Estimation module. Heterogeneous Division module projects representations into positive and negative spaces for HCorr. Then CoRA conducts H-PCorr Contrastive Learning in each space to guide projection and capture PCorr. (b) The DCorr Estimation module estimates correlations by combining Rule-based Correlations and Learnable Correlations, which are computed by Time-aware Polynomial and Time-Varying and Time-Invariant (T-T) Composition. (c) H-PCorr contrastive learning minimizes distances between strongly correlated channels and maximizes separation between weakly correlated channels in both positive and negative spaces.

In this work, we propose a **Cor**relation-aware **A**dapter (**CoRA**), a lightweight plugin that allows the TSFMs to capture various correlations during the fine-tuning stage. The framework of CoRA is visualized in Figure 2 . CoRA operates on input series, original predictions, and representations from TSFMs to enhance the prediction accuracy. Our method consists of four processes: **(i) Dynamic Correlation Estimation.** This module utilize representations from TSFMs and input series to learn dynamic correlations and generate correlation matrices that guide subsequent contrastive learning. **(ii) Heterogeneous Division.** To better capture HCorr, we design a channel-aware projection module that aggregates cross-channel context and adaptively adjusts channel-wise projection weights, thereby mapping the backbone representations into positive and negative latent spaces separately. **(iii) Heterogeneous Partial Correlation (H-PCorr) Contrastive Learning.** We propose H-PCorr Contrastive Learning within each representation of HCorr to learn PCorr by clustering only correlated channels. **(iv) Heterogeneous Fusion and Prediction.** Finally, we fuse the representations after contrastive learning for positive and negative correlations in Heterogeneous Fusion module and generate new predictions. Then, both original and new predictions are gated and added together.

## 4.1 DYNAMIC CORRELATION ESTIMATION

Channels exhibit both stable dependencies that do not change across time and fluctuations that change across time. Motivated by this, we introduce an innovative method that decomposes the learnable part of correlation matrix $\boldsymbol{M}_t^{corr} \in \mathbb{R}^{N \times N}$ at time $t$ into two low-rank components: Time-Varying $\boldsymbol{Q}_t \in \mathbb{R}^{N \times M}$ and Time-Invariant $\boldsymbol{V} \in \mathbb{R}^{M \times M}$, which can separate distinct correlation components, as illustrated in Figure 3. Here, $\boldsymbol{R} \in \mathbb{R}^{N \times N}$ denotes the rule-based correlation

matrix which is added to the learnable part to incorporate more prior knowledge for enhancing correlation estimation. $M$ is the hyperparameter for the post-decomposition rank, with $M < N$. This decomposition of the learnable part offers greater parameter efficiency, yet remains functionally equivalent to additive decomposition (Cirstea et al., 2021), as formally proven in Theorem 1.

We estimate the two components separately and then compose them back into the original correlation. The Time-Varying component represents the fluctuations in correlations across time. As time series data inherently have trends and periodic characteristics, the correlations that measure their dependencies also exhibit such variations across the entire time series (Liu et al., 2022). Thus, we propose Learnable Time-aware Polynomials to estimate the changes, as polynomials can be effective in modeling temporal patterns by sharing a common basis across different time steps. Based on a global adaptive method, the Time-Invariant component aims to capture the stable depen-

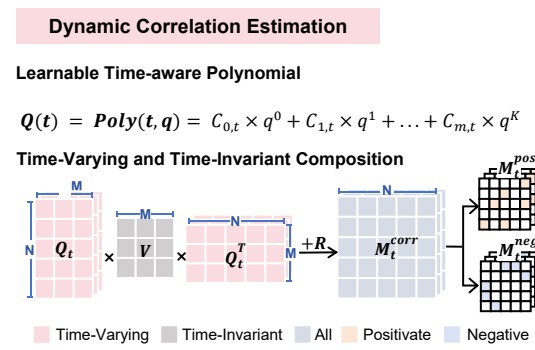

Figure 3: The details of DCorr Estimation

dencies among channels that do not change over time. Finally, we compute the correlation matrix $\boldsymbol{M}_t^{corr}$ by composing learnable correlations and combining with the rule-based correlation $\boldsymbol{R}$. This correlation matrix is then used for H-PCorr Contrastive Learning.

### 4.1.1 LEARNABLE TIME-AWARE POLYNOMIALS

Most existing approaches (Shang et al., 2024a; Cirstea et al., 2021; Zhao et al., 2023) struggle to accurately express the time-varying characteristics of DCorr due to the lack of explicit modeling of dynamic regularities.

In a stationary time series, we can use a well-behaved mathematical function to effectively approximate the fluctuations of the correlation. Considering that high-order polynomials provide better non-linear capacity than first-order ones, we use learnable polynomials to estimate $\boldsymbol{Q}_t$. The proof of this approximation capability is detailed in Theorem 2.

We construct a $K$-order Time-aware Polynomials with a shared matrix basis:

$$\boldsymbol{Q}_t = \sum_{i=0}^{K} C_{i,t} \boldsymbol{q}^i, \ (\boldsymbol{q}^i = \underbrace{\boldsymbol{q} \odot \boldsymbol{q} \odot \cdots \odot \boldsymbol{q}}_{i \text{ times}}) \ , \tag{1}$$

where $\boldsymbol{Q}_t \in \mathbb{R}^{N \times M}$ denote the Time-Varying component at time step $t$. $C_{i,t} \in \mathbb{R}^N$ is the $i$-th coefficient that varies over time, while $\boldsymbol{q} \in \mathbb{R}^{N \times M}$ is the globally learnable basis, which represents the pattern of changes over time. We define $\boldsymbol{q}^i$ as the $i$-times Hadamard product of the matrix $\boldsymbol{q}$, where the operation $\odot$ is the element-wise Hadamard product.

For convenience, we define the collection of $C_{i,t}$ as the matrix $\boldsymbol{\mathcal{C}}_t = (C_{0,t}, \cdots C_{K,t}) \in \mathbb{R}^{N \times K}$. It is the dependency coefficient of each channel for pattern $q^i$ and exhibits different values at different times, determined by specific data. Therefore, we learn the mapping $f$ between the representations of time series $\tilde{\boldsymbol{X}}_t$ and coefficients $\boldsymbol{\mathcal{C}}_t$ to estimate it :

$$\boldsymbol{\mathcal{C}}_t = f(\tilde{\boldsymbol{X}}_t) \in \mathbb{R}^{N \times K} \ . \tag{2}$$

Since only the polynomial coefficients need to be estimated with $f$, rather than the entire varying component, we can use a simple MLP to implement it.

### 4.1.2 TIME-VARYING AND TIME-INVARIANT COMPOSITION

Since the time-invariant part does not change over time, it should be globally unique. Inspired by self-learned graphs (Shang et al., 2024a; Wu et al., 2020), we use global learnable vectors to capture the implicit stable dependencies among channels:

$$\boldsymbol{V} = \text{Sigmoid}(\text{ReLU}(\boldsymbol{E}_1 \boldsymbol{E}_2^T)) \ , \tag{3}$$

where $\boldsymbol{V} \in \mathbb{R}^{M \times M}$ denote the Time-Invariant component of DCorr. $\boldsymbol{E}_1, \boldsymbol{E}_2 \in \mathbb{R}^{M \times d_e}$ are learnable vectors, $d_e$ is used to expand the dimensions, thereby enhancing the representation capacity.

As the statistics-based Pearson coefficient can describe simple linear correlation, we use it as the initialization for the final DCorr and build upon it to learn more complex correlations. The Pearson coefficient is calculated as follows:

$$r_{x,y} = \frac{\sum_{i=1}^{L}(x_i - \bar{x})(y_i - \bar{y})}{\sqrt{\sum_{i=1}^{L}(x_i - \bar{x})^2}\sqrt{\sum_{i=1}^{L}(y_i - \bar{y})^2}}, \tag{4}$$

where $x, y$ are two variables in $\boldsymbol{X}_t$; $r_{x,y}$ denotes the correlation coefficient among them, while $\bar{x}$ and $\bar{y}$ indicate the mean values of $x$ and $y$, respectively, $L$ is the size of input series. We use $\boldsymbol{R} \in \mathbb{R}^{N \times N}$ to denote the collection of $r$. Overall, our DCorr includes rule-based correlation $\boldsymbol{R}$, time-varying components $\boldsymbol{Q}_t$, and time-invariant components $\boldsymbol{V}$. The final estimated correlation is formulated as the sum of the learnable and rule-based parts in the following equation:

$$\boldsymbol{M}_t^{\text{corr}} = \boldsymbol{R} + \boldsymbol{Q}_t \boldsymbol{V} \boldsymbol{Q}_t^T. \tag{5}$$

## 4.2 Heterogeneous Correlation Division

Positive and negative correlations affect the channels differently, and the proportion of their contribution also varies among different channels. Motivated by Squeeze-and-Excitation (Hu et al., 2018) (SE), we propose a channel-aware projector that adjusts the weights of channels based on contextual information during projection to better project the representations into positive and negative spaces and learn Heterogeneous Correlation (HCorr).

To distinguish the dependency of channels on heterogeneous correlations, we project the representations into two latent spaces. Motivated by the SE mechanism, we design a channel-aware projection layer that aggregates cross-channel context and adaptively computes channel-wise projection weights. The channel-aware projection layer $\mathcal{P}$ with $\tilde{\boldsymbol{X}}_t^{\text{in}}$ and $\tilde{\boldsymbol{X}}_t^{\text{out}}$ is shown as follows:

$$\tilde{\boldsymbol{X}}_t^{\text{proj}} = g_1\Big(\text{LayerNorm}(\tilde{\boldsymbol{X}}_t^{\text{in}})\Big) \in \mathbb{R}^{P \times N \times d} \, , \;\; \boldsymbol{A}_t = \text{SoftMax}\Big(g_2(\tilde{\boldsymbol{X}}_t^{\text{proj}})\Big) \in \mathbb{R}^{P \times N \times 1} \, , \tag{6}$$

$$\tilde{\boldsymbol{X}}_t^{\text{ctx}} = \boldsymbol{A}_t^\top \tilde{\boldsymbol{X}}_t^{\text{proj}} \in \mathbb{R}^{P \times 1 \times d} \, , \;\; \tilde{\boldsymbol{X}}_t^{\text{fuse}} = g_3\Big([\tilde{\boldsymbol{X}}_t^{\text{proj}}, \text{expand}(\tilde{\boldsymbol{X}}_t^{\text{ctx}})]\Big) \in \mathbb{R}^{P \times N \times d} \, , \tag{7}$$

$$\boldsymbol{W}_t = \text{SoftMax}\Big(g_4\Big(\text{Pool}(\tilde{\boldsymbol{X}}_t^{\text{fuse}})\Big)\Big) \in \mathbb{R}^N \, , \;\; \tilde{\boldsymbol{X}}_t^{\text{out}} = \tilde{\boldsymbol{X}}_t^{\text{in}} + \tilde{\boldsymbol{X}}_t^{\text{fuse}} \odot \text{expand}(\boldsymbol{W}_t), \tag{8}$$

where $g_1$-$g_4$ are learnable multilayer perceptrons with mappings $\mathbb{R}^d \to \mathbb{R}^d$, $\mathbb{R}^d \to \mathbb{R}^1$, $\mathbb{R}^{2d} \to \mathbb{R}^d$, and $\mathbb{R}^d \to \mathbb{R}^1$, respectively; SoftMax$(\cdot)$ denotes softmax normalization along the channel dimension; $\boldsymbol{A}_t$ and $\boldsymbol{W}_t$ denote the patch-wise channel attention weights and adaptive channel-wise projection weights, respectively; Pool$(\cdot)$ denotes pooling over the patch dimension.

We perform two identical projection transformations on $\tilde{\boldsymbol{X}}_t$ to obtain the representations of the positive and negative latent spaces:

$$\tilde{\boldsymbol{X}}_t^{\text{pos}} = \mathcal{P}_1(\tilde{\boldsymbol{X}}_t) \in \mathbb{R}^{(P \times N \times d)}, \;\; \tilde{\boldsymbol{X}}_t^{\text{neg}} = \mathcal{P}_2(\tilde{\boldsymbol{X}}_t) \in \mathbb{R}^{(P \times N \times d)} \, , \tag{9}$$

where $\mathcal{P}_1$ and $\mathcal{P}_2$ are the proposed channel-aware projection operations with distinct learned parameters, and they are implemented as multi-layer architectures of $\mathcal{P}$. $\tilde{\boldsymbol{X}}_t^{\text{pos}}$ and $\tilde{\boldsymbol{X}}_t^{\text{neg}}$ are representations projected into spaces of positive and negative correlations, respectively. They contain channel information with adaptive adjustments and will subsequently be used for contrastive learning.

It is noteworthy that the Heterogeneous Correlation Division module cannot directly accomplish the disentanglement of heterogeneous correlations. Instead, this separation is achieved under the guidance of the contrastive learning framework detailed in the next section.

## 4.3 Heterogeneous Partial Correlation Contrastive Learning

To capture Partial Correlation, we design Partial Contrastive Learning, which uses the correlation matrix derived from Dynamic Correlation Estimation (DCE) and representations from Heterogeneous Correlation Division (HD) to enable adaptive correlation learning.

We leverage Contrastive Learning's advantages for clustering to capture PCorr (Yang et al., 2023). Compared to existing methods (Chen et al., 2024; Qiu et al., 2025d), this approach facilitates the fine-grained interaction among channels. Moreover, it does not add an extra burden during inference.

First based on the estimated correlation $M_t^{corr}$, we define the heterogeneous correlations as $M_t^{\text{pos}}$ and $M_t^{\text{neg}}$ to decouple the complex interactions among channels:

$$M_t^{\text{pos}} = \begin{cases} m_t^{\text{corr}}, & \text{if } corr > \epsilon \\ 0, & else \end{cases}, \quad M_t^{\text{neg}} = \begin{cases} m_t^{\text{corr}}, & \text{if } corr < -\epsilon \\ 0, & else \end{cases}, \quad (10)$$

where $m_t^{\text{corr}}$ is the element of $M_t^{\text{corr}}$, $\epsilon$ is the learnable threshold. We describe the process in the positive latent space; the negative latent space is handled analogously.

The matrix $M_t^{\text{pos}}$ is used to select positive and negative samples for each channel. In the designed contrastive learning if $M_t^{\text{pos}}[i, j] = 0$, it is considered a negative pair; otherwise, it is considered a positive pair. The loss for the positive correlation can be expressed as follows:

$$\mathcal{L}_{\text{pos}} = -\frac{1}{N} \sum_{i=1}^{N} log\left(\frac{\sum_{j=1}^{N} M_t^{\text{pos}}[i, j] \exp(\text{sim}(\tilde{X}_t^{\text{pos}}[i], \tilde{X}_t^{\text{pos}}[j])/\tau)}{\sum_{k=1}^{N} \exp(\text{sim}(\tilde{X}_t^{\text{pos}}[i], \tilde{X}_t^{\text{pos}}[k])/\tau)}\right), \quad (11)$$

where $\text{sim}(\cdot)$ represents the cosine similarity, and $\tau$ is the temperature coefficient used to control the degree of contrastive learning constraints. We do this for negative correlation in a similar way. The total auxiliary loss is defined as $\mathcal{L}_{\text{aux}} = \mathcal{L}_{\text{pos}} + \mathcal{L}_{\text{neg}}$.

### 4.4 HETEROGENEOUS FUSION AND PREDICTION

Finally, we project the representations of the two heterogeneous latent spaces into a shared space and then fuse them to perform prediction. Considering that some channels may require more correlation interaction while others may require more independence, we conduct a convex combination:

$$\tilde{X}_t^{\text{pos}} = \mathcal{P}_3(\tilde{X}_t^{\text{pos}}), \quad \tilde{X}_t^{\text{neg}} = \mathcal{P}_4(\tilde{X}_t^{\text{neg}}), \quad (12)$$

$$\hat{Y}_t^* = \beta \, \text{Linear}(\tilde{X}_t^{\text{pos}} + \tilde{X}_t^{\text{neg}}) + (1 - \beta) \, \hat{Y}_t, \quad (13)$$

where $\mathcal{P}_3$ and $\mathcal{P}_4$ consist of $l_2$ stacked channel-aware projection layers $\mathcal{P}$, as given by equations (6-8). Linear denotes the linear prediction head, $\beta \in [0, 1]^N$ is the learning gated weight.

### 4.5 COMPLEXITY ANALYSIS

The computational complexities are $\mathcal{O}(N^2)$ for the DCorr Estimation (DCE, Section 4.1) and H-PCorr Contrastive learning (HPCL, Section 4.3), and $\mathcal{O}(N)$ for HCorr Division (HD, Section 4.2). Most of the complexity arises from DCE and HPCL, which are only required during training. In the inference phase, since CoRA only includes HD modules, the time complexity is $\mathcal{O}(N)$.

Figure 5 shows CoRA imposes only minimal additional time on TSFMs, during fine-tuning and inference. The details of complexity analysis are included in Appendix B.

## 5 THEORETICAL ANALYSIS

### 5.1 THE SIGNIFICANCE OF TIME-VARYING AND TIME-INVARIANT COMPOSITION

A straightforward approach to modeling dynamic correlations is to decompose the correlation matrix into the sum of a time-varying matrix and a time-invariant matrix (Cirstea et al., 2021; Wu et al., 2019). However, this approach has parameter complexity. Our method can reduce the complexity while achieving the same effect.

**Theorem 1** *When the time series is locally stationary, the Time-Varying and Time-Invariant Decomposition allows $Q_t V Q_t^T$ to contain both time-varying and time-invariant information, like conventional additive decomposition.*

*Specifically, $Q_t V Q_t^T$ can be expressed as the sum of a time-invariant matrix $M_i$ and a time-varying matrix $M_v$, as shown below:*

$$Q_t V Q_t^T = M_i + M_v . \quad (14)$$

This indicates that our decomposition approach remains functionally equivalent to conventional additive decomposition. Notably, the expression $\boldsymbol{Q}_t \boldsymbol{V} \boldsymbol{Q}_t^T$ is the learnable component of the correlation matrix $\boldsymbol{M}_t^{corr}$, as defined in Equation 5.

## 5.2 THE FITTING ABILITY OF TIME-AWARE POLYNOMIALS

Time-aware polynomials can model complex time-varying correlation relationships, and the error bound decreases as the degree $K$ of the polynomial increases.

**Theorem 2** *When the time series is locally stationary, we can approximate the underlying correlation matrix with a high-order polynomial.*

*Specifically, assuming that the correlation is a smooth function of the basis $\boldsymbol{q}$, the true correlation component $\boldsymbol{Q}_t^*$ can be expressed as $\mathcal{F}(\boldsymbol{q})$. The error bound can be formalized as follows.*

$$|\boldsymbol{Q_t^*} - \boldsymbol{Q_t}| = \frac{\mathcal{F}^{(K+1)}(\boldsymbol{\xi})}{(K+1)!} \boldsymbol{q}^{(K+1)}, \boldsymbol{\xi} \in [-|\boldsymbol{q}|, |\boldsymbol{q}|] . \tag{15}$$

This indicates that by selecting an appropriate $K$, we can strike an effective balance between model effectiveness and computational efficiency, thereby enabling the efficient estimation of dynamic correlations. We provide the proof of Theorems 1-2 in the Appendix C.

## 6 EXPERIMENT

### 6.1 EXPERIMENTAL DETAILS

**Datasets.** To conduct comprehensive and fair comparisons for different models, we conduct experiments on ten well-known forecasting benchmarks as the target datasets, including ETT (4 subsets), Electricity, Traffic, Solar, weather, AQShunyi and ZafNoo, which cover multiple domains. More details of the benchmark datasets are included in Table 4 of Appendix D.1.

**Baselines and Implementation.** We choose the latest state-of-the-art models to serve as baselines, including 3 Time Series LLM-based models (GPT4TS, CALF, UniTime) and 3 Time Series pre-trained models (Moment, Chronos, Timer). We utilize the TSFM-Bench (Li et al., 2025) code repository for unified evaluation. More implementation details are included in D.3. To keep consistent with previous works, we adopt Mean Squared Error (MSE) and Mean Absolute Error (MAE) as evaluation metrics. We provide our code at https://github.com/decisionintelligence/CoRA.

### 6.2 MAIN RESULTS

Comprehensive forecasting results of TSFMs with and without using CoRA are listed in Table 1. The look-back window length ($L$) is set to 576 for Timer and 512 for the other models. We have the following observations: i) Compared to fine-tuning without CoRA, fine-tuning with CoRA achieves better results in average results and results of different forecasting horizons (Table 6 and Table 7, in Appendix E) for both LLM-based models and time series pre-trained models, even in 5% few-shot settings. ii) Sharing the same pre-trained parameters, TTM's Channel-Dependent (CD) and Channel-Independent (CI) versions differ only in their module configurations during downstream fine-tuning. We implement a CI version of TTM, fine-tuned with CoRA and compare it with a CD version which is fine-tuned without CoRA. The better performance of the former demonstrates that considering the mentioned three types of correlations allows the model to better understand the inter-channels interaction.

### 6.3 COMPARISON WITH OTHER CORRELATION PLUGINS

To better validate the effectiveness of CoRA, we compare it with LIFT Zhao & Shen (2024) and C-LoRA Nie et al. (2024). We select GPT4TS, UniTime and Timer as the backbone and set H to 96. As shown in Figure 4, since LIFT and C-LoRA are not specifically designed for TSFMs, the limited training samples in the few-shot setting lead to a degradation in their performance, negatively impacting the effectiveness of the TSFMs. In contrast, CoRA, designed specifically for TSFMs, learns multiple correlations from the representations, allowing it to fully leverage their capabilities.

Table 1: Multivariate forecasting results in the 5% few-shot setting, where MSE is averaged over four forecasting horizons $H \in \{96, 192, 336, 720\}$. The better results are highlighted in **bold**. Full and MAE results are available in Appendix E.

| Model | LLM-based | | | | | | Pre-trained | | | | | |
|---|---|---|---|---|---|---|---|---|---|---|---|---|
| | GPT4TS (2023) | | CALF (2025) | | UniTime (2024) | | Moment (2024) | | Timer (2024) | | TTM (2024) | |
| Plugin | ✗ | ✓ | ✗ | ✓ | ✗ | ✓ | ✗ | ✓ | ✗ | ✓ | ✗ | ✓ |
| ETTh1 | 0.464 | **0.453** | 0.453 | **0.445** | 0.721 | **0.682** | 0.438 | **0.433** | 0.450 | **0.421** | 0.397 | **0.392** |
| ETTh2 | 0.374 | **0.369** | 0.367 | **0.362** | 0.406 | **0.392** | 0.351 | **0.348** | 0.380 | **0.361** | 0.340 | **0.335** |
| ETTm1 | 0.387 | **0.369** | 0.373 | **0.363** | 0.405 | **0.381** | 0.358 | **0.351** | 0.443 | **0.426** | 0.358 | **0.352** |
| ETTm2 | 0.274 | **0.268** | 0.276 | **0.269** | 0.272 | **0.262** | 0.260 | **0.255** | 0.277 | **0.272** | 0.260 | **0.256** |
| Electricity | 0.208 | **0.201** | 0.175 | **0.169** | 0.201 | **0.194** | 0.202 | **0.199** | 0.241 | **0.236** | 0.179 | **0.173** |
| Traffic | 0.441 | **0.431** | 0.429 | **0.421** | 0.456 | **0.446** | 0.451 | **0.444** | 0.456 | **0.447** | 0.485 | **0.436** |
| Solar | 0.256 | **0.245** | 0.213 | **0.194** | 0.253 | **0.248** | 0.227 | **0.223** | 0.217 | **0.213** | 0.219 | **0.197** |
| Weather | 0.254 | **0.245** | 0.236 | **0.229** | 0.255 | **0.244** | 0.252 | **0.249** | 0.241 | **0.237** | 0.226 | **0.222** |
| AQShunyi | 0.717 | **0.712** | 0.734 | **0.727** | 0.738 | **0.716** | 0.694 | **0.681** | 0.734 | **0.724** | 0.703 | **0.693** |
| ZafNoo | 0.544 | **0.525** | 0.544 | **0.541** | 0.618 | **0.605** | 0.531 | **0.526** | 0.538 | **0.524** | 0.504 | **0.498** |

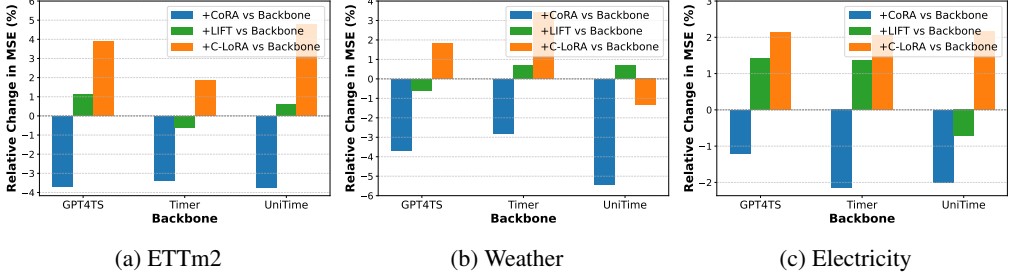

(a) ETTm2         (b) Weather         (c) Electricity

Figure 4: Correlation Plugins comparison in 5% few-shot fine-tuning setting.

## 6.4 ABLATION STUDIES

To investigate the effectiveness of CoRA, we conduct comprehensive experiments. In our work, the DCorr Estimation (DCE) is used to learn DCorr and generate the labels required for H-PCorr Contrastive Learning (HPCL), while HPCL utilizes these labels to guide the projectors in the Heterogeneous Division (HD) module. Therefore, they cannot operate independently. We utilize naive implementations in place of the original modules within certain variants.

Table 2: The MSE results of various variants.

| | Dataset | | | ETTm2 | | | Electricity | | |
|---|---|---|---|---|---|---|---|---|---|
| | DCorr Estimation | HCorr Dvision | H-PCorr Contrastive Learning | GPT4TS | UniTime | Timer | GPT4TS | UniTime | Timer |
| 1 | ✗ | ✗ | ✗ | 0.274 | 0.272 | 0.277 | 0.208 | 0.201 | 0.241 |
| 2 | ⭕ | ⭕ | ✓ | 0.273 | 0.270 | 0.275 | 0.206 | 0.199 | 0.239 |
| 3 | ⭕ | ✓ | ✓ | 0.271 | 0.268 | 0.274 | 0.204 | 0.197 | 0.239 |
| 4 | ✓ | ⭕ | ✓ | 0.270 | 0.269 | 0.273 | 0.205 | 0.198 | 0.238 |
| 5 | ✓ | ✓ | ✓ | **0.268** | **0.262** | **0.257** | **0.201** | **0.194** | **0.236** |

✗ denotes a module removed, ✓ denotes a module added, ⭕ denotes replace a module with a naive implementation.

Specifically, we replace the DCE module with a series-level Pearson correlation coefficient, which cannot model DCorr, and the HD module with a single-branch projection layer, which is unable to capture PCorr. The comparison between Row 1-2 demonstrates the effectiveness of HPCL; however, its performance is limited due to its inability to capture multiple types of correlations. In Rows 2-4, the addition of either the DCE or HD module to HPCL further enhances performance, which confirms the efficacy of both modules. In Row 5, the combination of all three modules achieves the best performance.

## 6.5 MODEL ANALYSIS

**Efficiency Analysis** Our proposed CoRA, as a lightweight plugin for TSFMs, shows strong efficiency, particularly during the inference phase. Figure 5 shows a comparative analysis of the efficiency of TSFMs with and without the application of CoRA. We selected three datasets in ascending order of the number of channels: ETTm2 ($N = 7$), Weather ($N = 21$), and Electricity ($N = 321$). For the experiments, both the look-back window $L$ and the forecasting horizon $H$ were set to 96. Train time and Inference time refer to the duration of a single training epoch and the total time required to process all samples during inference, respectively. The results show that, compared to the backbone itself, the use of CoRA does not introduce significant additional time or parameter numbers. Moreover, as the number of channels ($N$) increases from 7, 21, to 321, CoRA maintains its efficiency without noticeable degradation compared to the backbone, particularly during inference.

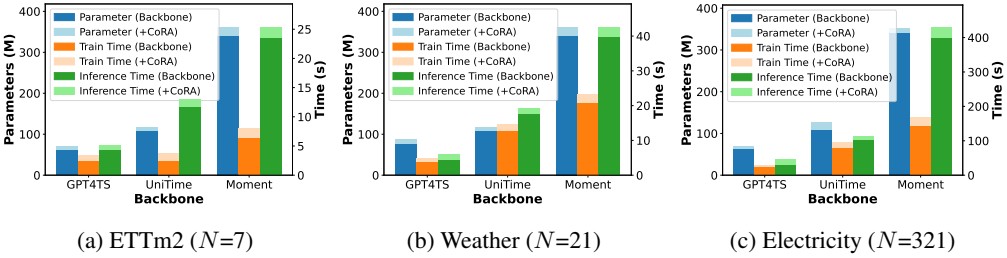

(a) ETTm2 ($N$=7)     (b) Weather ($N$=21)     (c) Electricity ($N$=321)

Figure 5: Efficiency Analysis of TSFMs with and without the application of CoRA.

**Data Analysis** While the previous results focused on the 5% fine-tuning setting, we expand the analysis to provide a more comprehensive view and to explicitly explore the impact of fine-tuning data volume on performance. Specifically, we fine-tune the TTM and CALF backbones on the ETTm2 and Weather datasets, using 3%, 5%, 10%, and 20% of the available training data. The MSE results

Table 3: MSE results for different data percentage.

| Dataset | ETTm2 | | | | Weather | | | |
|---|---|---|---|---|---|---|---|---|
| Data | 3% | 5% | 10% | 20% | 3% | 5% | 10% | 20% |
| **TTM** | 0.264 | 0.260 | 0.255 | 0.250 | 0.239 | 0.227 | 0.223 | 0.218 |
| | ±.006 | ±.004 | ±.003 | ±.003 | ±.003 | ±.002 | ±.003 | ±.004 |
| **+ CoRA** | **0.260** | **0.255** | **0.248** | **0.245** | **0.235** | **0.222** | **0.212** | **0.209** |
| | ±.004 | ±.003 | ±.002 | ±.002 | ±.004 | ±.003 | ±.003 | ±.002 |
| **CALF** | 0.284 | 0.277 | 0.268 | 0.260 | 0.252 | 0.237 | 0.230 | 0.226 |
| | ±.004 | ±.003 | ±.003 | ±.003 | ±.005 | ±.003 | ±.003 | ±.004 |
| **+ CoRA** | **0.280** | **0.267** | **0.259** | **0.255** | **0.247** | **0.230** | **0.225** | **0.218** |
| | ±.003 | ±.002 | ±.002 | ±.002 | ±.003 | ±.002 | ±.002 | ±.004 |

are summarized in the Table 3. As the results indicate, CoRA still yields a modest performance improvement even in a low-data regime using only 3% of the data.

**Sensitivity Analysis and Visualization** The Prarmeter Sensitivity analyses for the polynomial's degree $K$, the decomposition size $M$, and the number of projection layers $l_1, l_2$ are presented in Appendix F.1. The Visualization of heterogeneous spaces are presented in Appendix F.2.

## 7 CONCLUSION

In this paper, we propose a lightweight Correlation-aware Adapter (CoRA) that enhances the predictive performance of Time Series Foundation Models (TSFMs) by considering the mentioned three types of correlation relationships. We combine Time-Varying and Time-Invariant Composition with Learnable Time-Aware Polynomials to enable lightweight modelling of dynamic correlations exhibiting regular patterns. For Heterogeneous Correlation and Partial Correlation, we design channel-aware heterogeneous projections together with Heterogeneous-Partial Contrastive Learning. This approach decouples Heterogeneous Correlation while simultaneously achieving adaptive Partial Correlation learning among channels. Comprehensive experiments on real-world datasets demonstrate that CoRA can improve the forecast performance of TSFMs.

ACKNOWLEDGEMENTS

This work was partially supported by the National Natural Science Foundation of China (No. 62372179, No. 62472174) and the Fundamental Research Funds for the Central Universities. Chenjuan Guo is the corresponding author of the work.

ETHICS STATEMENT

Our work exclusively uses publicly available benchmark datasets that contain no personally identifiable information. The proposed adapter for Time Series Foundation Models in Multivariate Time Series Forecasting is designed for beneficial applications in system reliability and safety monitoring. No human subjects were involved in this research.

REPRODUCIBILITY STATEMENT

The performance of CoRA and the datasets used in our work are real, and all experimental results can be reproduced. We have released our model code at: https://github.com/decisionintelligence/CoRA.

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

# A    DEFINITIONS OF THE THREE CORRELATIONS

**Definition 1** (*Correlation Martix*) Given a multivariate time series $X \in \mathbb{R}^{N \times L}$, where $N$ represents the number of channels and $L$ the temporal length, the series is partitioned into a set of $K = \lfloor L/T \rfloor$ non-overlapping segments using a window of length $T$. For the $k$-th segment, we define the channel-wise correlation matrix as $C^{(k)} \in \mathbb{R}^{N \times N}$, where $k \in [1, K]$. The element at the i-th row and j-th column of this matrix is denoted by $C_{ij}^{(k)}$ which represents the correlation between the i-th channel and the j-th channel.

**Definition 2** (*Dynamic Correlation*) The *Dynamic Correlation* refers to the case where the correlation martix $C^{(k)}(k \in [1, K])$ is not constant over the segments index $k$. Formally, this means there exist at least two distinct segment indices $m, n \in [1, K]$ with $m \neq n$, such that their corresponding correlation matrices are unequal: $C^{(m)} \neq C^{(n)}$.

**Definition 3** (*Heterogeneous Correlation*) The *Heterogeneous Correlation* refers to the case where there exists at least one temporal segment $k \in [1, K]$ in which some channels exhibit both positive and negative correlations with other channels. Formally, this means that for at least one correlation matrix $C^{(k)}(k \in [1, K])$, there exist at least three distinct channel indices $a, b, c \in [1, N]$, such that the corresponding correlation matrix elements $c_{ab}^{(k)}$ and $c_{ac}^{(k)}$ have opposite signs: $c_{ab}^{(k)} \cdot c_{ac}^{(k)} < 0$.

**Definition 4** (*Partial Correlation*) The *Partial Correlation* refers to the case where, within at least one temporal segment $k \in [1, K]$, there exists some pairs of channels with a non-significant relationship. Formally, given a predefined significance threshold $\epsilon$, this means there exists at least one matrix $C^{(k)}(k \in [1, K])$ and at least one pair of distinct channel indices $a, b \in [1, N]$, such that: $|c_{ab}^{(k)}| < \epsilon$.

# B    COMPLEXITY ANALYSES

## B.1    TRAINING PHASE

**Dynamic Correlation Estimation** This module consits of Learnable Time-aware Polynomials (LTP) and Time-Varying and Time-Invariant (T-T) Composition. The LTP have a computational complexity of $\mathcal{O}(PKNM + PNd^2)$ due to the polynomial operations and the MLP used to generate $\mathcal{C}_t$, and a space complexity of $\mathcal{O}(Kd + MN)$ because of the basis $q$ and the MLP used to generate $\mathcal{C}_t$. Where $P$ is the number of patches, $N$ denotes the number of channels, $M$ is the second dimension of $Q_t$, $K$ is the degree of the polynomial and $d$ is the dimension of representations. The T-T Composition has a computational complexity of $\mathcal{O}(lN^2 + NM^2 + MN^2)$ due to the calculation of the Pearson coefficient and the composition in Equation 5. Where $l$ denotes the patch size. **Heterogeneous Division** This module has a computational complexity of $\mathcal{O}(PNd^2)$ and a space complexity of $\mathcal{O}(PNd)$ due to the channel-aware projection. **H-PCorr Contrastive Learning** The time complexity of calculating loss in Equation 11 is $\mathcal{O}(PdN^2)$.

Since $P$, $K$, $M$, and $l$ are much smaller than $N$ and $d$, they will not be considered as the primary components in the complexity analysis. So the total computational complexity is $\mathcal{O}(dN^2 + Nd^2)$ and the total space complexity is $\mathcal{O}(d + N)$. Most models cannot avoid having a computational complexity of $\mathcal{O}(d^2)$ and a space complexity of $\mathcal{O}(d)$. Therefore, if we focus only on the complexity with respect to (N) in our discussion, our model has a computational complexity of $\mathcal{O}(N^2)$ and a space complexity of $\mathcal{O}(N)$ during training.

## B.2    INFERENCE PHASE

In inference Phase, CoRA only includes projectors in the Heterogeneous Division and Heterogeneous Fusion modules, based on the above discussion, our model has a computational complexity of $\mathcal{O}(N)$ and a space complexity of $\mathcal{O}(1)$ during inference.

## C  THEORETICAL ANALYSES

### C.1  THE SIGNIFICANCE OF TIME-VARYING AND TIME-INVARIANT COMPOSITION

The channel correlation can be expressed by combining a long-term stable state with dynamic changes. We decompose the learnable correlation into two parts, $Q_t$ and $V$, as shown in Figure 3, to fit the correlation relationship lightweightly.

**Theorem 1** *When the time series is locally stationary, the Time-Varying and Time-Invariant Decomposition allows $Q_t V Q_t^T$ to contain both time-varying and time-invariant information, like conventional additive decomposition.*

*Specifically, $Q_t V Q_t^T$ can be expressed as the sum of a time-invariant matrix $M_i$ and a time-varying matrix $M_v$, as shown below:*

$$Q_t V Q_t^T = M_i + M_v \ . \tag{16}$$

*Proof.* Under the assumptions of locally stationary, $Q_t$ can be expressed as $\bar{Q}_t + \tilde{Q}_t$, where $\bar{Q}_t$ represents the mean value and $\tilde{Q}_t$ represents the residual. Therefore, $M_t^{corr}$ can be expressed as:

$$
\begin{aligned}
Q_t V Q_t^T &= (\bar{Q}_t + \tilde{Q}_t) V (\bar{Q}_t + \tilde{Q}_t)^T \\
&= \bar{Q}_t V \bar{Q}_t^T + \bar{Q}_t V \tilde{Q}_t^T + \tilde{Q}_t V \bar{Q}_t^T + \tilde{Q}_t V \tilde{Q}_t^T \\
&= (\bar{Q}_t V \bar{Q}_t^T) + (\bar{Q}_t V \tilde{Q}_t^T + \tilde{Q}_t V \bar{Q}_t^T + \tilde{Q}_t V \tilde{Q}_t^T) \\
&= M_i + M_v \ ,
\end{aligned}
\tag{17}
$$

where $M_i = \bar{Q}_t V \bar{Q}_t^T$ has the same value at different times, and $M_v = \bar{Q}_t V \tilde{Q}_t^T + \tilde{Q}_t V \bar{Q}_t^T + \tilde{Q}_t V \tilde{Q}_t^T$ has different values at different times.

### C.2  THE FITTING ABILITY OF TIME-AWARE POLYNOMIALS

Since time series exhibit regular changes, such as trends and seasonality, the dynamic correlation changes also have a certain regularity. To this end, we propose Time-aware Polynomials to fit the changing correlations better.

**Theorem 2** *When the time series is locally stationary, we can approximate the underlying correlation matrix with a high-order polynomial.*

*Specifically, assuming that the correlation is a smooth function to the basis $q$, the true correlation component $Q_t^*$ can be expressed as $\mathcal{F}(q)$. The fitting error of Time-aware Polynomials decreases as the highest degree $K$ of the polynomial increases. The error can be formalized as follows:*

$$|Q_t^* - Q_t| = \frac{\mathcal{F}^{(K+1)}(\xi)}{(K+1)!} q^{(K+1)}, \xi \in [-|q|, |q|] \ . \tag{18}$$

*Proof.* Given the true correlation as $\mathcal{F}(q)$, Since $\mathcal{F}$ is sufficiently smooth to the basis $q$, we can perform a Maclaurin expansion of $\mathcal{F}$ around $0$:

$$\mathcal{F}(q) = \mathcal{F}(0) + \mathcal{F}'(0)q + \frac{\mathcal{F}''(0)}{2!} q^2 + \cdots + \frac{\mathcal{F}^{(K)}(0)}{K!} q^K + \cdots + \frac{\mathcal{F}^{(n)}(0)}{n!} q^n + \cdots \ . \tag{19}$$

We construct auxiliary functions:

$$\mathcal{H}(t) = \mathcal{F}(q) - [\mathcal{F}(t) + \mathcal{F}'(t)(q - t) + \frac{\mathcal{F}''(t)}{2!}(q - t)^2 + \cdots + \frac{\mathcal{F}^{(K)}(t)}{K!}(q - t)^K] \ , \tag{20}$$

$$\mathcal{G}(t) = (q - t)^{(K+1)} \ . \tag{21}$$

Assume $q > 0$. Then, $\mathcal{H}$ and $\mathcal{G}$ are still continuously differentiable, and the following rules apply:

$$\mathcal{H}(t)' = -\frac{\mathcal{F}^{(K+1)}(t)}{K!}(q - t)^K \ , \tag{22}$$

$$\mathcal{G}(t)' = -(K+1)(q - t)^K \neq 0 \ . \tag{23}$$

Since $\mathcal{H}(\boldsymbol{q}) = \mathcal{G}(\boldsymbol{q}) = 0$, by the Cauchy Mean Value Theorem, $\exists\, \boldsymbol{\xi} \in (0, \boldsymbol{q}), s.t.$

$$\frac{\mathcal{H}(\mathbf{0})}{\mathcal{G}(\mathbf{0})} = \frac{\mathcal{H}(\mathbf{0}) - \mathcal{H}(\boldsymbol{q})}{\mathcal{G}(\mathbf{0}) - \mathcal{G}(\boldsymbol{q})} = \frac{\mathcal{H}(\mathbf{0}) - \mathcal{H}(\boldsymbol{q})}{\mathcal{G}(\mathbf{0}) - \mathcal{G}(\boldsymbol{q})} = \frac{\mathcal{H}'(\boldsymbol{\xi})}{\mathcal{G}'(\boldsymbol{\xi})} = \frac{\mathcal{F}^{(K+1)}(\boldsymbol{\xi})}{(K+1)!} \ . \tag{24}$$

Therefore, we can derive the following equation:

$$\mathcal{F}(\boldsymbol{q}) = \mathcal{F}(\mathbf{0}) + \mathcal{F}'(\mathbf{0})\boldsymbol{q} + \frac{\mathcal{F}''(\mathbf{0})}{2!}\boldsymbol{q}^2 + \cdots + \frac{\mathcal{F}^{(K)}(\mathbf{0})}{K!}\boldsymbol{q}^K + \frac{\mathcal{F}^{(K+1)}(\boldsymbol{\xi})}{(K+1)!}\boldsymbol{q}^{(K+1)}, \boldsymbol{\xi} \in [0, \boldsymbol{q}] \ . \tag{25}$$

Let $C_{i,t} = \frac{\mathcal{F}^{(i)}}{i!}$. Then, we have the following equation:

$$\mathcal{F}(\boldsymbol{q}) = C_{0,t} + C_{1,t}\boldsymbol{q} + C_{2,t}\boldsymbol{q}^2 + \cdots + C_{K,t}\boldsymbol{q}^K + \frac{\mathcal{F}^{(K+1)}(\boldsymbol{\xi})}{(K+1)!}\boldsymbol{q}^{(K+1)}, \boldsymbol{\xi} \in [0, \boldsymbol{q}] \ . \tag{26}$$

That is:

$$\mathcal{F}(\boldsymbol{q}) - \boldsymbol{Q}_t = \frac{\mathcal{F}^{(K+1)}(\boldsymbol{\xi})}{(K+1)!}\boldsymbol{q}^{(K+1)}, \boldsymbol{\xi} \in [0, \boldsymbol{q}] \ . \tag{27}$$

For all $\boldsymbol{q} > 0$ and $\boldsymbol{q} < 0$, we have the following equation:

$$|\mathcal{F}(\boldsymbol{q}) - \boldsymbol{Q}_t| = \frac{\mathcal{F}^{(K+1)}(\boldsymbol{\xi})}{(K+1)!}\boldsymbol{q}^{(K+1)}, \boldsymbol{\xi} \in [-|\boldsymbol{q}|, |\boldsymbol{q}|] \ . \tag{28}$$

And that is:

$$|\boldsymbol{Q}_t^* - \boldsymbol{Q}_t| = \frac{\mathcal{F}^{(K+1)}(\boldsymbol{\xi})}{(K+1)!}\boldsymbol{q}^{(K+1)}, \boldsymbol{\xi} \in [-|\boldsymbol{q}|, |\boldsymbol{q}|] \ . \tag{29}$$

# D EXPERIMENTAL DETAILS

## D.1 DATASETS

To conduct comprehensive and fair comparisons for different models, we conduct experiments on ten well-known forecasting benchmarks as the target datasets, including: (I) **ETT** (Zhou et al., 2021) datasets contain 7 variates collected from two different electric transformers from July 2016 to July 2018. It consists of four subsets, of which ETTh1/ETTh2 are recorded hourly, and ETTm1/ETTm2 are recorded every 15 minutes. (II) **Electricity** (Trindade, 2015) contains the electricity consumption of 321 customers from July 2016 to July 2019, recorded hourly. (III) **Traffic** (Wu et al., 2021) contains road occupancy rates measured by 862 sensors on freeways in the San Francisco Bay Area from 2015 to 2016, recorded hourly. (IV) **Solar** (Lai et al., 2018) records solar power generation from 137 PV plants in 2006, every 10 minutes. (V) **Weather** (Wu et al., 2021) collects 21 meteorological indicators, including temperature and barometric pressure, for Germany in 2020, recorded every 10 minutes. (VI) **AQShunyi** (Zhang et al., 2017) is an air quality dataset from a measurement station, for 4 years. (VII) **ZafNoo** (Poyatos et al., 2020) is collected from the Sapflux data project and includes sap flow measurements and environmental variables. The details of the benchmark datasets are included in Table 4

Table 4: Statistics of datasets.

| Dataset | Domain | Frequency | Lengths | Dim | Split | Description |
|---|---|---|---|---|---|---|
| ETTh1 | Electricity | 1 hour | 14,400 | 7 | 6:2:2 | Power transformer 1, comprising seven indicators such as oil temperature and useful load |
| ETTh2 | Electricity | 1 hour | 14,400 | 7 | 6:2:2 | Power transformer 2, comprising seven indicators such as oil temperature and useful load |
| ETTm1 | Electricity | 15 mins | 57,600 | 7 | 6:2:2 | Power transformer 1, comprising seven indicators such as oil temperature and useful load |
| ETTm2 | Electricity | 15 mins | 57,600 | 7 | 6:2:2 | Power transformer 2, comprising seven indicators such as oil temperature and useful load |
| Weather | Environment | 10 mins | 52,696 | 21 | 7:1:2 | Recorded every for the whole year 2020, which contains 21 meteorological indicators |
| Electricity | Electricity | 1 hour | 26,304 | 321 | 7:1:2 | Electricity records the electricity consumption in kWh every 1 hour from 2012 to 2014 |
| Solar | Energy | 10 mins | 52,560 | 137 | 6:2:2 | Solar production records collected from 137 PV plants in Alabama |
| Traffic | Traffic | 1 hour | 17,544 | 862 | 7:1:2 | Road occupancy rates measured by 862 sensors on San Francisco Bay area freeways |
| AQShunyi | Environment | 1 hour | 35,064 | 11 | 7:1:2 | Air quality dataset from a measurement station, for 4 years |
| ZafNoo | Nature | 30 mins | 19,225 | 11 | 7:1:2 | Sap flow measurements and environmental variables from the Sapflux data project. |

## D.2 BASELINES

Remarkable achievements in deep learning have been witnessed across time series analysis (Wang et al., 2026a; Tian et al., 2026; Yu et al., 2025b; Wang et al., 2026b; Cheng et al., 2026; Wang et al., 2025c; Yu et al., 2025a; Wang et al., 2025b; Shang et al., 2026; Wang et al., 2025a; Shang et al., 2024b; Wang et al., 2023; Chen et al., 2023), computer vision (Ma et al., 2024a; 2025a), and various other domains(Ma et al., 2025b; Shao et al., 2025; Tian et al., 2026; Wu et al., 2026; Li et al., 2026; Ma et al., 2014). Existing literature indicates that features extracted automatically usually surpass human-designed ones in terms of performance(Ma et al., 2024b; Wu et al., 2025b; Ma et al., 2025c; Wu et al., 2025a; Liu et al., 2026a; Wang et al., 2026c; Cheng et al., 2023).

In the realm of time series forecasting, numerous models have surfaced in recent years. We choose models with superior predictive performance in our benchmark, including the pre-trained time series models: Timer (Liu et al., 2024e), TTM (Ekambaram et al., 2024b) and Moment (Goswami et al., 2024); The LLM-based models: CALF (Liu et al., 2025), GPT4TS (Zhou et al., 2023), UniTime (Liu et al., 2024b); The specific descriptions for each of these models—see Table 5.

Table 5: Descriptions of time series forecasting models in experiment.

| Models | Descriptions |
|---|---|
| Moment (Goswami et al., 2024) | Moment is a transformer system pre-trained on a masked time series task. It reconstructs masked portions of time series for tasks like forecasting, classification, anomaly detection, and imputation. |
| TTM (Ekambaram et al., 2024b) | It is based on MLP-Mixer blocks with gated attention and multi-resolution sampling. It captures temporal patterns and cross-channel correlations for time-series forecasting, optimized for zero/few-shot learning with low computational cost. |
| Timer (Liu et al., 2024e) | Timer is a GPT-style autoregressive model for time series analysis, predicting the next token in single-series sequences. It supports tasks like forecasting, imputation, and anomaly detection across different time series. |
| CALF (Liu et al., 2025) | CALF is a cross-modal knowledge distillation framework that aligns time series data with pre-trained LLMs by leveraging both static and dynamic knowledge, achieving state-of-the-art performance in both long- and short-term forecasting tasks with strong generalization abilities. |
| GPT4TS (Zhou et al., 2023) | GPT4TS fine-tunes the limited parameters of LLM, which demonstrates competitive performance by transferring knowledge from large-scale pre-training text data. |
| UniTime (Liu et al., 2024b) | UniTime designs domain instructions to align time series and text modalities. |

### D.3 IMPLEMENTATION DETAILS

We utilize the TSFM-Bench (Li et al., 2025) code repository for unified evaluation. Following the settings in TFB (Qiu et al., 2024) and FMTS-Bench (Li et al., 2025), we do not apply the Drop Last trick to ensure a fair comparison. All experiments of CoRA are conducted using PyTorch in Python 3.10. The MSE loss function guides the training process and employs the ADAM optimizer.

## E  FULL RESULTS

Table 6: The table reports MSE and MAE of LLM-based models for different forecasting horizons $F \in \{96, 192, 336, 720\}$. The better results are highlighted in **bold**.

| Model | | GPT4TS (2023) | | | | CALF (2025) | | | | UniTime (2024) | | | |
|---|---|---|---|---|---|---|---|---|---|---|---|---|---|
| Plugin | | ✗ | | ✓ | | ✗ | | ✓ | | ✗ | | ✓ | |
| Metric | | MSE | MAE | MSE | MAE | MSE | MAE | MSE | MAE | MSE | MAE | MSE | MAE |
| ETTh1 | 96 | 0.436 | 0.444 | **0.425** | **0.432** | 0.414 | 0.425 | **0.410** | **0.423** | 0.709 | 0.579 | **0.667** | **0.567** |
| | 192 | 0.453 | 0.455 | **0.436** | **0.442** | 0.442 | 0.441 | **0.438** | **0.439** | 0.715 | 0.585 | **0.671** | **0.553** |
| | 336 | 0.461 | 0.467 | **0.449** | **0.452** | 0.460 | 0.455 | **0.456** | **0.454** | 0.712 | 0.594 | **0.677** | **0.576** |
| | 720 | 0.507 | 0.510 | **0.501** | **0.508** | 0.496 | 0.494 | **0.474** | **0.481** | 0.749 | 0.630 | **0.713** | **0.602** |
| ETTh2 | 96 | 0.321 | 0.376 | **0.309** | **0.360** | 0.308 | 0.364 | **0.300** | **0.357** | 0.374 | 0.420 | **0.359** | **0.403** |
| | 192 | 0.368 | 0.406 | **0.365** | **0.398** | 0.362 | 0.396 | **0.358** | **0.394** | 0.389 | 0.431 | **0.372** | **0.415** |
| | 336 | 0.378 | 0.421 | **0.377** | **0.417** | 0.381 | 0.415 | **0.380** | **0.413** | 0.387 | 0.431 | **0.380** | **0.429** |
| | 720 | 0.427 | 0.458 | **0.423** | **0.456** | 0.417 | 0.445 | **0.411** | **0.440** | 0.474 | 0.490 | **0.455** | **0.457** |
| ETTm1 | 96 | 0.343 | 0.380 | **0.334** | **0.374** | 0.313 | 0.364 | **0.302** | **0.357** | 0.357 | 0.386 | **0.342** | **0.358** |
| | 192 | 0.369 | 0.394 | **0.353** | **0.383** | 0.348 | 0.383 | **0.340** | **0.377** | 0.387 | 0.402 | **0.377** | **0.378** |
| | 336 | 0.393 | 0.406 | **0.371** | **0.389** | 0.382 | 0.402 | **0.376** | **0.399** | 0.414 | 0.417 | **0.379** | **0.395** |
| | 720 | 0.441 | 0.434 | **0.417** | **0.417** | 0.447 | 0.437 | **0.432** | **0.430** | 0.460 | 0.441 | **0.424** | **0.425** |
| ETTm2 | 96 | 0.189 | 0.280 | **0.184** | **0.274** | 0.180 | 0.270 | **0.176** | **0.267** | 0.188 | 0.273 | **0.170** | **0.260** |
| | 192 | 0.241 | 0.312 | **0.237** | **0.307** | 0.243 | 0.311 | **0.237** | **0.307** | 0.239 | 0.306 | **0.230** | **0.300** |
| | 336 | 0.294 | 0.349 | **0.283** | **0.334** | 0.291 | 0.341 | **0.287** | **0.340** | 0.289 | 0.338 | **0.281** | **0.333** |
| | 720 | 0.370 | 0.391 | **0.367** | **0.388** | 0.389 | 0.405 | **0.374** | **0.393** | 0.372 | 0.391 | **0.368** | **0.389** |
| Electricity | 96 | 0.180 | 0.295 | **0.173** | **0.288** | 0.143 | 0.244 | **0.139** | **0.238** | 0.174 | 0.283 | **0.168** | **0.277** |
| | 192 | 0.194 | 0.308 | **0.186** | **0.302** | 0.160 | 0.259 | **0.155** | **0.254** | 0.186 | 0.292 | **0.181** | **0.285** |
| | 336 | 0.209 | 0.319 | **0.204** | **0.314** | 0.178 | 0.277 | **0.173** | **0.271** | 0.201 | 0.305 | **0.195** | **0.300** |
| | 720 | 0.248 | 0.349 | **0.241** | **0.340** | 0.220 | 0.312 | **0.210** | **0.302** | 0.242 | 0.337 | **0.233** | **0.325** |
| Traffic | 96 | 0.440 | 0.323 | **0.429** | **0.314** | 0.403 | 0.296 | **0.395** | **0.289** | 0.424 | 0.311 | **0.414** | **0.303** |
| | 192 | 0.423 | 0.305 | **0.413** | **0.299** | 0.418 | 0.303 | **0.411** | **0.297** | 0.437 | 0.320 | **0.427** | **0.309** |
| | 336 | 0.433 | 0.309 | **0.425** | **0.301** | 0.427 | 0.309 | **0.421** | **0.303** | 0.476 | 0.333 | **0.465** | **0.325** |
| | 720 | 0.468 | 0.325 | **0.455** | **0.315** | 0.466 | 0.329 | **0.455** | **0.322** | 0.485 | 0.363 | **0.477** | **0.354** |
| Solar | 96 | 0.243 | 0.262 | **0.233** | **0.251** | 0.184 | 0.245 | **0.177** | **0.236** | 0.251 | 0.286 | **0.246** | **0.278** |
| | 192 | 0.259 | 0.295 | **0.250** | **0.282** | 0.208 | 0.265 | **0.190** | **0.250** | 0.251 | 0.321 | **0.246** | **0.313** |
| | 336 | 0.260 | 0.278 | **0.250** | **0.267** | 0.225 | 0.281 | **0.201** | **0.258** | 0.253 | 0.323 | **0.247** | **0.316** |
| | 720 | 0.260 | 0.281 | **0.248** | **0.275** | 0.235 | 0.294 | **0.206** | **0.263** | 0.256 | 0.327 | **0.253** | **0.320** |
| Weather | 96 | 0.189 | 0.244 | **0.181** | **0.237** | 0.163 | 0.216 | **0.157** | **0.210** | 0.184 | 0.240 | **0.175** | **0.229** |
| | 192 | 0.226 | 0.276 | **0.218** | **0.265** | 0.206 | 0.254 | **0.199** | **0.251** | 0.228 | 0.275 | **0.217** | **0.265** |
| | 336 | 0.270 | 0.306 | **0.260** | **0.294** | 0.252 | 0.289 | **0.245** | **0.286** | 0.271 | 0.305 | **0.259** | **0.293** |
| | 720 | 0.330 | 0.348 | **0.320** | **0.336** | 0.322 | 0.340 | **0.315** | **0.333** | 0.335 | 0.351 | **0.323** | **0.336** |
| AQShunyi | 96 | 0.668 | 0.494 | **0.657** | **0.487** | 0.694 | 0.511 | **0.681** | **0.502** | 0.695 | 0.508 | **0.658** | **0.484** |
| | 192 | 0.702 | 0.509 | **0.699** | **0.503** | 0.720 | 0.521 | **0.713** | **0.519** | 0.725 | 0.520 | **0.702** | **0.507** |
| | 336 | 0.721 | 0.520 | **0.719** | **0.518** | 0.735 | 0.528 | **0.730** | **0.525** | 0.741 | 0.530 | **0.724** | **0.520** |
| | 720 | 0.775 | 0.546 | **0.774** | **0.539** | 0.788 | 0.552 | **0.782** | **0.550** | 0.791 | 0.552 | **0.780** | **0.543** |
| ZafNoo | 96 | 0.493 | 0.473 | **0.476** | **0.467** | 0.478 | 0.443 | **0.475** | **0.442** | 0.557 | 0.528 | **0.544** | **0.519** |
| | 192 | 0.530 | 0.488 | **0.502** | **0.476** | 0.531 | 0.474 | **0.528** | **0.471** | 0.606 | 0.555 | **0.587** | **0.534** |
| | 336 | 0.561 | 0.501 | **0.540** | **0.472** | 0.566 | 0.492 | **0.564** | **0.488** | 0.640 | 0.571 | **0.631** | **0.564** |
| | 720 | 0.591 | 0.518 | **0.582** | **0.493** | 0.602 | 0.514 | **0.596** | **0.510** | 0.670 | 0.586 | **0.659** | **0.575** |

Table 7: The table reports MSE and MAE of pre-trained models for different forecasting horizons $F \in \{96, 192, 336, 720\}$. The better results are highlighted in **bold**.

| Model | | Moment (2024) | | | | Timer (2024) | | | | TTM (2024) | | | |
|---|---|---|---|---|---|---|---|---|---|---|---|---|---|
| CoRA | | ✗ | | ✓ | | ✗ | | ✓ | | ✗ | | ✓ | |
| Metric | | MSE | MAE | MSE | MAE | MSE | MAE | MSE | MAE | MSE | MAE | MSE | MAE |
| ETTh1 | 96 | 0.400 | 0.416 | **0.394** | **0.414** | 0.383 | 0.408 | **0.363** | **0.396** | 0.363 | 0.392 | **0.360** | **0.389** |
| | 192 | 0.423 | 0.431 | **0.420** | **0.428** | 0.450 | 0.448 | **0.409** | **0.427** | 0.393 | 0.411 | **0.386** | **0.406** |
| | 336 | 0.436 | 0.448 | **0.429** | **0.443** | 0.446 | 0.450 | **0.418** | **0.441** | 0.405 | 0.424 | **0.401** | **0.420** |
| | 720 | 0.494 | 0.492 | **0.489** | **0.488** | 0.520 | 0.517 | **0.493** | **0.492** | 0.428 | 0.453 | **0.421** | **0.449** |
| ETTh2 | 96 | 0.294 | 0.354 | **0.290** | **0.349** | 0.292 | 0.341 | **0.277** | **0.333** | 0.273 | 0.331 | **0.270** | **0.329** |
| | 192 | 0.351 | 0.391 | **0.348** | **0.388** | 0.370 | 0.400 | **0.357** | **0.393** | 0.336 | 0.371 | **0.331** | **0.366** |
| | 336 | 0.364 | 0.408 | **0.360** | **0.403** | 0.421 | 0.455 | **0.393** | **0.435** | 0.367 | 0.398 | **0.355** | **0.394** |
| | 720 | 0.395 | 0.436 | **0.393** | **0.433** | 0.436 | 0.464 | **0.416** | **0.446** | 0.385 | 0.427 | **0.382** | **0.423** |
| ETTm1 | 96 | 0.306 | 0.354 | **0.301** | **0.349** | 0.286 | 0.341 | **0.281** | **0.338** | 0.302 | 0.346 | **0.291** | **0.339** |
| | 192 | 0.339 | 0.371 | **0.335** | **0.365** | 0.354 | 0.387 | **0.347** | **0.382** | 0.343 | 0.369 | **0.336** | **0.365** |
| | 336 | 0.368 | 0.387 | **0.360** | **0.383** | 0.382 | 0.406 | **0.380** | **0.403** | 0.367 | 0.382 | **0.362** | **0.377** |
| | 720 | 0.418 | 0.416 | **0.406** | **0.411** | 0.749 | 0.560 | **0.695** | **0.541** | 0.421 | 0.413 | **0.419** | **0.412** |
| ETTm2 | 96 | 0.176 | 0.265 | **0.167** | **0.260** | 0.167 | 0.248 | **0.163** | **0.247** | 0.165 | 0.252 | **0.163** | **0.250** |
| | 192 | 0.226 | 0.299 | **0.224** | **0.296** | 0.237 | 0.301 | **0.235** | **0.299** | 0.223 | 0.291 | **0.221** | **0.288** |
| | 336 | 0.278 | 0.333 | **0.273** | **0.332** | 0.307 | 0.355 | **0.297** | **0.344** | 0.279 | 0.330 | **0.275** | **0.323** |
| | 720 | 0.360 | 0.387 | **0.356** | **0.384** | 0.395 | 0.413 | **0.394** | **0.408** | 0.372 | 0.386 | **0.364** | **0.381** |
| Electricity | 96 | 0.140 | 0.236 | **0.137** | **0.231** | 0.159 | 0.242 | **0.158** | **0.235** | 0.146 | 0.246 | **0.141** | **0.240** |
| | 192 | 0.163 | 0.257 | **0.161** | **0.253** | 0.187 | 0.265 | **0.183** | **0.256** | 0.165 | 0.264 | **0.159** | **0.258** |
| | 336 | 0.185 | 0.281 | **0.180** | **0.278** | 0.258 | 0.287 | **0.256** | **0.285** | 0.181 | 0.281 | **0.175** | **0.274** |
| | 720 | 0.320 | 0.368 | **0.316** | **0.359** | 0.360 | 0.372 | **0.348** | **0.364** | 0.223 | 0.315 | **0.215** | **0.308** |
| Traffic | 96 | 0.383 | 0.272 | **0.379** | **0.266** | 0.392 | 0.278 | **0.390** | **0.272** | 0.448 | 0.324 | **0.410** | **0.299** |
| | 192 | 0.414 | 0.286 | **0.411** | **0.280** | 0.444 | 0.297 | **0.429** | **0.287** | 0.466 | 0.330 | **0.427** | **0.306** |
| | 336 | 0.436 | 0.299 | **0.426** | **0.294** | 0.436 | 0.302 | **0.429** | **0.293** | 0.491 | 0.345 | **0.435** | **0.309** |
| | 720 | 0.571 | 0.484 | **0.559** | **0.477** | 0.552 | 0.494 | **0.540** | **0.478** | 0.533 | 0.365 | **0.471** | **0.326** |
| Solar | 96 | 0.170 | 0.219 | **0.169** | **0.216** | 0.203 | 0.270 | **0.201** | **0.268** | 0.201 | 0.254 | **0.183** | **0.248** |
| | 192 | 0.197 | 0.248 | **0.196** | **0.241** | 0.215 | 0.276 | **0.209** | **0.274** | 0.225 | 0.270 | **0.195** | **0.260** |
| | 336 | 0.203 | 0.255 | **0.198** | **0.254** | 0.225 | 0.282 | **0.224** | **0.272** | 0.222 | 0.274 | **0.197** | **0.260** |
| | 720 | 0.336 | 0.337 | **0.328** | **0.335** | 0.225 | 0.281 | **0.218** | **0.277** | 0.226 | 0.277 | **0.214** | **0.270** |
| Weather | 96 | 0.150 | 0.200 | **0.148** | **0.195** | 0.170 | 0.227 | **0.164** | **0.224** | 0.148 | 0.195 | **0.145** | **0.193** |
| | 192 | 0.215 | 0.264 | **0.214** | **0.259** | 0.211 | 0.260 | **0.208** | **0.258** | 0.194 | 0.238 | **0.189** | **0.234** |
| | 336 | 0.283 | 0.317 | **0.278** | **0.312** | 0.256 | 0.293 | **0.251** | **0.283** | 0.247 | 0.280 | **0.243** | **0.277** |
| | 720 | 0.360 | 0.375 | **0.356** | **0.366** | 0.327 | 0.343 | **0.323** | **0.337** | 0.315 | 0.330 | **0.312** | **0.327** |
| AQShunyi | 96 | 0.534 | 0.434 | **0.524** | **0.432** | 0.729 | 0.491 | **0.728** | **0.475** | 0.645 | 0.481 | **0.637** | **0.478** |
| | 192 | 0.713 | 0.521 | **0.705** | **0.511** | 0.706 | 0.509 | **0.685** | **0.495** | 0.691 | 0.507 | **0.684** | **0.498** |
| | 336 | 0.735 | 0.526 | **0.715** | **0.518** | 0.724 | 0.521 | **0.709** | **0.520** | 0.707 | 0.515 | **0.693** | **0.504** |
| | 720 | 0.792 | 0.543 | **0.779** | **0.529** | 0.778 | 0.544 | **0.774** | **0.528** | 0.767 | 0.541 | **0.757** | **0.532** |
| ZafNoo | 96 | 0.437 | 0.400 | **0.431** | **0.396** | 0.476 | 0.442 | **0.475** | **0.430** | 0.428 | 0.405 | **0.422** | **0.399** |
| | 192 | 0.522 | 0.453 | **0.517** | **0.452** | 0.522 | 0.465 | **0.504** | **0.448** | 0.487 | 0.441 | **0.480** | **0.433** |
| | 336 | 0.548 | 0.480 | **0.547** | **0.471** | 0.558 | 0.482 | **0.540** | **0.476** | 0.528 | 0.461 | **0.523** | **0.450** |
| | 720 | 0.617 | 0.514 | **0.610** | **0.508** | 0.594 | 0.501 | **0.575** | **0.489** | 0.573 | 0.489 | **0.565** | **0.484** |

# F    MORE ANALYSIS ON CORA

## F.1    HYPERPARAMETER SENSITIVITY

With TTM as the backbone and H set to 96, we study the hyperparameter sensitivity of CoRA, including the Degree of Polynomial ($K$), the size of decomposition ($M$), the layers' number of projectors before and after HPCL ($l_1$ and $l_2$). Figure 6a show that $K$ is a robust hyperparameter, and we often choose 3 or 4 as common configurations. Figure 6b illustrates that the selection of M does not need to increase rapidly with the number of channels. Figure 6c and Figure 6d show that too few layers may lead to insufficient fitting capacity, while too many can diminish generalization ability. We often choose 3 or 5 as common configurations.

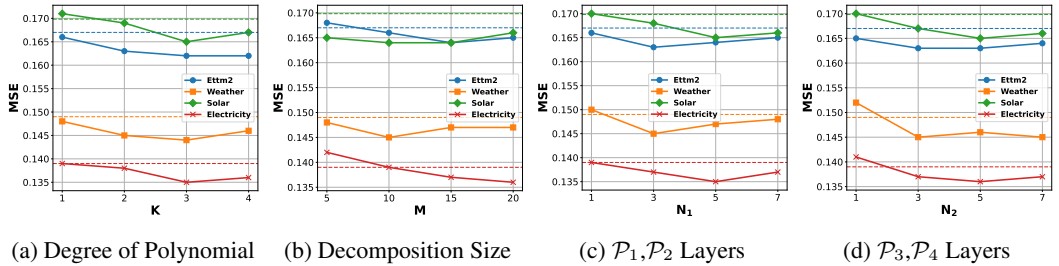

(a) Degree of Polynomial      (b) Decomposition Size      (c) $\mathcal{P}_1, \mathcal{P}_2$ Layers      (d) $\mathcal{P}_3, \mathcal{P}_4$ Layers

Figure 6: Parameter sensitivity of main hyper-parameters in CoRA.

## F.2    VISUALIZATION OF HETEROGENEOUS SPACES.

To further demonstrate the effectiveness of CoRA in modeling the DCorr, HCorr and PCorr, we conduct a visualization experiment. Specifically, by examining samples at 3 time steps and 4 channels in the Weather dataset, we compare the similarities between representations in the heterogeneous space. The result is shown in Figure 7. Among them, Figure 7a illustrates the visualization of samples from the Weather dataset, with each time step comprising 64 time points. Figure 7b shows the cosine similarity between the channel representations in positive and negative spaces. Based on observations, it can be concluded that within the 0-64 time points, Channel 1 and Channel 3 as well as Channel 2 and Channel 4 exhibit significant positive correlations. Within the 64-128 time points, Channel 2 Channel 3 and Channel 4 show significant positive correlations, while Channel 3 demonstrates channel independence. Within the 128-192 time points, Channel 1 and Channel 3 exhibit significant positive correlations, while they show significant negative correlations with Channel 2 and Channel 4. These findings align with the actual data, demonstrating that CoRA is capable of simultaneously capturing DCorr HCorr and PCorr.

# G    THE USE OF LARGE LANGUAGE MODELS

The use of open-source Large Language Models (LLMs) in this work was strictly limited to assisting with the translation of certain terms and polishing a small portion of the text. LLMs did not contribute to the conceptual aspects of the research, including information retrieval, knowledge discovery, or the ideation process.

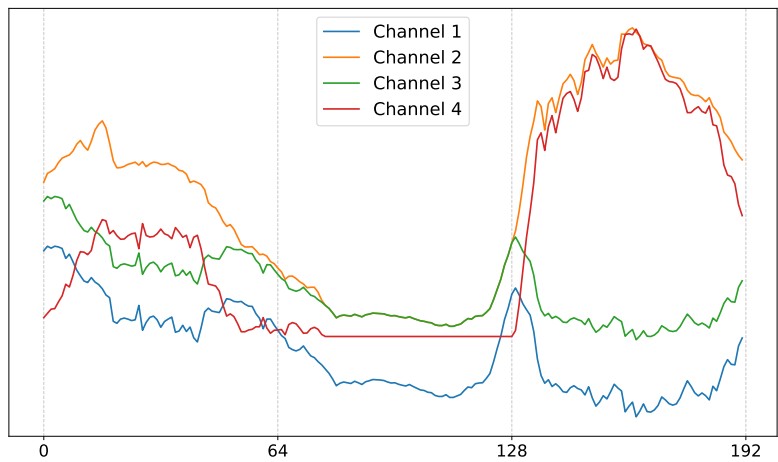

(a) Samples at 3 time steps and 4 channels in Weather dataset.

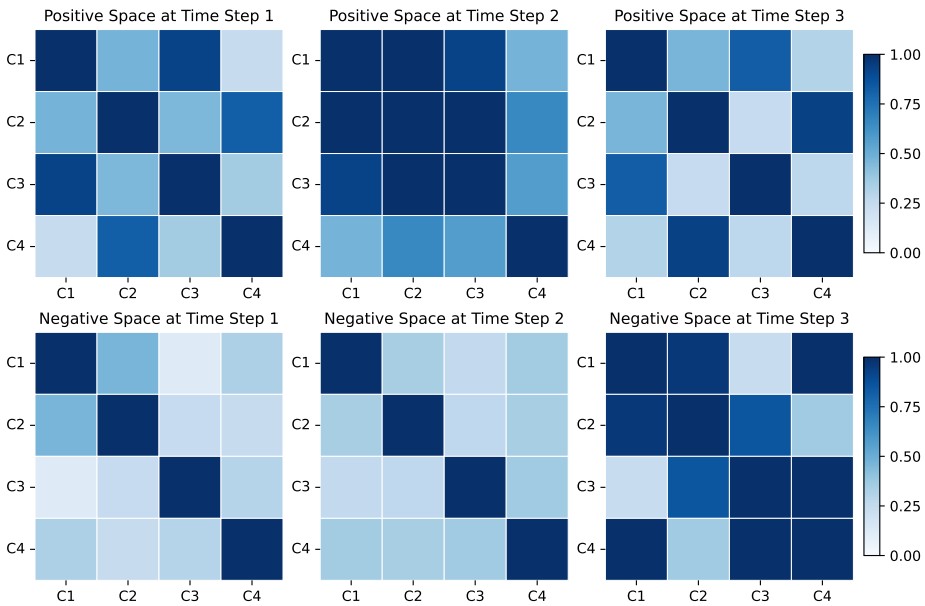

(b) The similarity of representations in positive and negative spaces at 3 time steps.

Figure 7: Visualization of Heterogeneous Spaces

