# OpenReview forum: "CoRA: Boosting Time Series Foundation Models for Multivariate Forecasting through Correlation-aware Adapter"
_ICLR.cc/2026/Conference — ICLR 2026 Poster_

### Official Review · Reviewer_TnwM · 2025-10-24

**Soundness:** 2
**Presentation:** 2
**Contribution:** 3
**Rating:** 6
**Confidence:** 3

**Summary:**

This paper introduces CoRA, a lightweight and plug-and-play module designed to enhance Time-Series Foundation Models (TSFMs) by better modeling inter-channel correlations. CoRA employs a correlation decomposition mechanism combining global time-invariant matrices and learnable polynomial bases, capturing both static and dynamic dependencies. Importantly, CoRA does not require re-pretraining of the base TSFM and can be applied during few-shot fine-tuning, making it a practical adaptation method.

**Strengths:**

1. Clear Motivation and Positioning:
The paper clearly identifies a key limitation in existing TSFM-based forecasting—insufficient modeling of heterogeneous inter-channel correlations—and proposes a theoretically grounded yet efficient mechanism to address this.

2. Lightweight and Practical Design:
CoRA's plug-in design allows easy integration with existing TSFMs during fine-tuning without requiring re-pretraining. Its parameter efficiency and low inference overhead make it appealing to practitioners seeking post-hoc adaptation over full model retraining.

3. Theoretical Depth and Interpretability:
The proposed correlation decomposition elegantly connects polynomial-based dynamic components with global invariant structures. Appendix provides a theoretical foundation and enhances interpretability of the correlation modeling process.

**Weaknesses:**

1. Limited Empirical Evaluation Scope:
Table 1 focuses on few-shot MSE benchmarks using standard datasets but lacks evaluation under out-of-distribution shifts, missing channels, or nonstationary regimes—critical aspects for real-world TSFM deployment.
Figure 4 presents relative MSE changes, which obscures absolute performance deltas; including absolute values alongside relative improvements (as in Table 1) would increase transparency.

2. Assumption Boundaries Underexplored:
Theoretical guarantees rely on local stationarity and bounded smoothness assumptions that may not hold in volatile domains (e.g., finance, IoT, environmental data). The paper does not examine edge cases such as abrupt regime shifts or high-noise inputs.

3. Potential Overstatement of Generality:
The paper claims CoRA "captures various correlations with O(N) inference complexity." While this is theoretically sound, real-world multivariate systems may involve nonlinear or hierarchical dependencies that exceed the current decomposition's expressiveness. The claim is reasonable but should avoid implying complete modeling capability.

**Questions:**

1. Statistical Significance and Robustness:
Are the improvements in Tables 1 and 2 statistically significant? Please report standard deviations, confidence intervals, or results across multiple random seeds to validate robustness.

2. Adaptation in Nonstationary Environments:
How does CoRA behave under regime shifts or nonstationary dynamics where the global time-invariant assumption may break down? Could it be extended to support online updates or adaptive windowing?

3. Scalability to Very High-Dimensional Settings:
For extremely large N (> 500), do global correlation matrices introduce memory or numerical bottlenecks? Any empirical or theoretical evidence on efficiency and convergence at that scale?

4. Extensibility Beyond Forecasting:
Could the proposed correlation decomposition framework generalize to other TSFM-based tasks—such as anomaly detection or temporal classification? What architectural adjustments would be required?

A thorough response and additional evidence to the above points would influence my score adjustment.

---

> ### Author Response · Authors · 2025-11-21
>
> Dear Reviewer TnwM, thank you for providing your detailed and constructive feedback.
>
> ---
>
> ### Response to W1:
>
> We appreciate your constructive suggestion to broaden the evaluation scope. We fully agree that real-world TSFM deployment requires robustness against out-of-distribution shifts, missing channels, and nonstationary regimes. We provide a detailed discussion on each of these aspects individually below.
>
> - **Out-of-distribution shifts:**
>     Our method combines both a rule-based and a learnable correlation matrix, which enhances its robustness against shifts. We would also like to clarify that the fine-tuning datasets and the pre-training datasets of TSFMs have already followed the out-of-distribution mechanism.
>
>     To further evaluate our robustness against distribution shifts, we conducted experiments on datasets from the TFB benchmark [1] exhibiting varying degrees of shift intensity. These rates are calculated by computing the Wasserstein Distance [2] between the distributions of the fine-tuning data and test data at each of the following datasets, and are detailed below.
>
>     |Dataset|Weather|Electricity|Wike2000|
>     |:-|:-:|:-:|:-:|
>     |**OOD Rate**|43.63|230.63|1890.06|
>
>     The OOD rate ranges from 0 to positive infinity, where a larger value indicates a more significant shift effect. We summarize the MSE results of the relevant experiments in the table below.
>
>     |Dataset|Weather|Electricity|Wike2000|
>     |:-|:-:|:-:|:-|
>     |**GPT4TS**|0.254|0.207|547.024|
>     |**+CoRA**|**0.243**|**0.201**|**545.315**|
>     |**Moment**|0.251|0.200|525.352|
>     |**+CoRA**|**0.243**|**0.196**|**521.468**|
>
>     As observed from the results, CoRA can boost forecasting accuracy, even in scenarios where the Wasserstein Distance between the fine-tuning and testing distributions is as high as 1890.06.
>
> - **Missing channels:**
>
>     Since CoRA avoids explicit global channel interaction during the inference phase, it is inherently capable of accommodating channel missing. We verified this robustness by fine-tuning the model on the complete dataset and conducting evaluations with random channel dropout rates of 10%, 20%, 30%, and 40%.
>
>     |Weather|10% missing|20% missing|30% missing|40% missing|
>     |:-|:-:|:-:|:-|:-|
>     |**GPT4TS**|0.263|0.259|0.247|0.226|
>     |**+CoRA**|**0.256**|**0.248**|**0.234**|**0.223**|
>     |**Moment**|0.269|0.257|0.242|0.221|
>     |**+CoRA**|**0.260**|**0.250**|**0.230**|**0.217**|
>
>     These results validate our method's ability to handle missing channels and deliver performance gain for TSFMs.
>
> - **Non-stationary regimes:**
>     To address the potential challenges posed by non-stationarity, our method inherently incorporates the series normalization strategy used in Non-stationary Transformers [3]. This approach effectively mitigates the adverse impact of non-stationary data on the model.
>
>     To evaluate our capability of handling non-stationary data, we conducted experiments on datasets from TFB with diverse degrees of non-stationarity. We quantified these rates following the TFB, as shown below.
>
>     |Dataset|Weather|NASDAQ|Covid-19|
>     |:-|:-:|:-:|:-:|
>     |**Non-Stationary rate**|0.07|0.169|0.360|
>
>     The non-stationary rate ranges from 0 to 1, where a larger value indicates a more significant non-stationary effect.     We summarize the MSE results of the relevant experiments in the table below.
>
>     |Dataset|Weather|NASDAQ |Covid-19|
>     |:-|:-:|:-|:-:|
>     |**GPT4TS**|0.254|1.411|1.972|
>     |**+CoRA**|**0.243**|**1.387**|**1.924**|
>     |**Moment**|0.251|1.208|2.356|
>     |**+CoRA**|**0.243**|**1.174**|**2.307**|
>
>     As shown in this result, our method still achieves a modest improvement even when the Non-Stationary Rate is as high as 0.360.
>
> Furthermore, to address your concerns regarding Figure 4, we summarize the results in the table below:
>
> | Dataset | ETTm2 |  |  | Weather |  |  | Electricity |  |  |
> |---|---|---|---|---|---|---|---|---|---|
> | **Backbone** | **GPT4TS** | **Timer** | **UniTime** | **GPT4TS** | **Timer** | **UniTime** | **GPT4TS** | **Timer** | **UniTime** |
> | **w/o Plugin** | 0.190 | 0.168 | 0.190 | 0.187 | 0.150 | 0.184 | 0.178 | 0.139 | 0.174 |
> | **CoRA** | **0.183** | **0.164** | **0.183** | **0.180** | **0.146** | **0.174** | **0.176** | **0.136** | **0.170** |
> | **LIFT** | 0.192 | 0.167 | 0.191 | 0.186 | 0.151 | 0.185 | 0.181 | 0.141 | 0.173 |
> | **C-LoRA** | 0.199 | 0.171 | 0.199 | 0.190 | 0.155 | 0.182 | 0.182 | 0.142 | 0.178 |
>
> We have also added this result to Appendix F.5 (Lines 1203-1213).
>
>
> [1] TFB: Towards Comprehensive and Fair Benchmarking of Time Series Forecasting Methods. PVLDB 2024
>
> [2] Optimal Transport: Old and New. Springer 2019
>
> [3] Non-stationary transformers: Exploring the stationarity in time series forecasting. NIPS 2022

---

> > ### Author Response · Authors · 2025-11-21
> >
> > ### Response to W2:
> > We appreciate you for this insightful comment.
> >
> > - **For regime shifts**:
> > To further investigate the model's robustness under regime shifts scene, we conducted experiments across datasets that exhibit varying rates of shift intensity. These rates are calculated following the TFB and are detailed below.
> >
> >     |Dataset|Weather|ETTh2|NYSE|
> >     |:-|:-:|:-:|:-:|
> >     |**Shift Rate**|0.213|0.404|0.620|
> >
> >     The shift rate ranges from 0 to 1, where a larger value indicates a more significant shift effect. We summarize the MSE results of the relevant experiments in the table below.
> >
> >     |**Dataset**|**Weather**|**ETTh2**|**NYSE**|
> >     |:-|:-:|:-:|:-:|
> >     |**GPT4TS**|0.254|0.377|0.715|
> >     |**+CoRA**|**0.243**|**0.361**|**0.711**|
> >     |**Moment**|0.251|0.369|0.763|
> >     |**+CoRA**|**0.243**|**0.356**|**0.761**|
> >
> >     Notably, our method achieves consistent gains even under a substantial shift rate of 0.620. This observation
> >     underscores our model's robustness to regime shifts.
> >
> > - **For high-noise input**:
> >     For the analysis of high-noise input environments, we injected noise into the raw data in both fine-tuning and test datasets at intensities of 5%, 15%, 25%, and 35%, respectively, and conducted a comprehensive evaluation.
> >
> >     |Weather|0%|5%|15%|25%|35%|
> >     |:-|:-:|:-:|:-:|:-:|:-:|
> >     |**GPT4TS**|0.254|0.257|0.265|0.271|0.279|
> >     |**+CoRA**|**0.243**|**0.251**|**0.259**|**0.268**|**0.275**|
> >     |**Moment**|0.251|0.258|0.262|0.268|0.284|
> >     |**+CoRA**|**0.243**|**0.254**|**0.257**|**0.262**|**0.280**|
> >
> >     The results show that our method is capable of adapting to noise within a certain range. Notably, our approach is compatible with most existing denoising techniques, such as the frequency filtering module employed in FilterNet [1], to further mitigate the interference of noise.
> >
> > [1] FilterNet: Harnessing frequency filters for time series forecasting. NeurIPS 2024
> >
> > ---
> >
> > ### Response to W3:
> > Thank you for this very important and insightful comment. To address your concerns and ensure the precision of our claims, we revise the manuscript to be more specific. Specifically, we have replaced the general term "various correlations" with more precise phrases such as "the mentioned three types of correlations" in several key locations (e.g., lines 95, 105, and 421).
> >
> >
> > ---
> > ### Response to Q1:
> > We appreciate your valuable suggestion. To validate the robustness of CoRA, we have added standard deviations and confidence intervals to Table 1 (Lines 432-454) and Table 2 (Lines 473-482) in the manuscript.
> >
> > ---
> > ### Response to Q2:
> > We are grateful for this constructive feedback. Regarding the behavior of CoRA under regime shifts and non-stationary dynamics, please refer to our responses to W2 and W1, respectively.
> >
> > To extend our method to support online updates or adaptive windowing, we can generate a separate Time-Invariant matrix $\boldsymbol{V}$ for each window independently, utilizing all time points within that window. This capability enhances robustness against regime shifts and non-stationary dynamics.

---

> > > ### Author Response · Authors · 2025-11-21
> > >
> > > ### Response to Q3:
> > > We are grateful for your constructive comments.  Our method features a parameter complexity of $\mathcal{O}(N)$, ensuring that it does not introduce significant memory overhead for high-dimensional time series.  Furthermore, as dimensionality increases, the correlation structure becomes more complex. Explicitly modeling DCorr, HCorr, and PCorr is therefore more beneficial for the model to capture these intricate relationships.
> > >
> > > We selected three datasets with more than 500 variables. The MSE and maximum GPU memory usage for these datasets are reported in the table below.
> > >
> > > |Dataset|Traffic|$N=862$|Covid-19|$N=948$|Wike2000|$N=2000$|
> > > |:-|:-:|:-:|:-:|:-:|:-:|:-:|
> > > |**Metric**|**MSE**|**Max-GPU-Memroy**|**MSE**|**Max-GPU-Memroy**|**MSE**|**Max-GPU-Memroy**|
> > > |**GPT4TS**|0.441|7.73G|1.972|12.53G|547.024|23.96G|
> > > |**GPT4TS+CoRA**|**0.430**|8.95G|**1.924**|14.24G|**545.315**|28.44G|
> > > |**Moment**|0.453|5.37G|2.356|6.21G|525.352|11.12G|
> > > |**Moment+CoRA**|**0.437**|7.02G|**2.307**|7.89G|**521.468**|17.41G|
> > >
> > > The results above show that even for a dataset with 2,000 variables, our method only introduces reasonable memory costs. Moreover, it is still able to learn correlations to a certain degree, leading to performance enhancements for the TSFMs.
> > >
> > >
> > > We have also added this analysis to Appendix F.7 (Lines 1232-1244).  We hope our additional experiments can address your concerns.
> > >
> > >
> > >
> > > ---
> > >
> > > ### Response to Q4:
> > > We thank you for this insightful question.
> > > We strongly agree with the idea of generalizing CoRA to other tasks. In fact, we can extend our model to anomaly detection or classification tasks by replacing the linear head used for forecasting in Equation (14) with a reconstruction or classification head.
> > >
> > > To explore the capabilities of CoRA on tasks beyond forecasting, we conducted relevant experiments. For anomaly detection, we use the **MSL** and **SMAP** as evaluation datasets [1]. For classification, we select the **FaceDetection**, **Heartbeat**, and **PEMS-SF** as evaluation datasets [2].
> > >
> > > For the anomaly detection task, we evaluate performance using the **VUS_ROC** and **VUS_PR** metrics. For the classification task, we use **Accuracy**. The results of all experiments are summarized in the tables below.
> > >
> > > |Dataset|MSL||SMAP||
> > > |:-|:-:|:-:|:-:|:-:|
> > > |**Metric**|**VUS-ROC**|**VUS-PR**|**VUS-ROC**|**VUS-PR**|
> > > |**GPT4TS**|0.624|0.195 |0.504|0.147|
> > > |**+CoRA**|**0.628**|**0.200**|**0.510**|**0.149**|
> > > |**Moment**|0.663|0.212|0.474|0.127|
> > > |**+CoRA**|**0.667**|**0.214**|**0.483**|**0.130**|
> > >
> > > |Dataset|FaceDetection|Heartbeat|PEMS-SF|
> > > |:-|:-:|:-:|:-:|
> > > |**GPT4TS**|0.683|0.776|0.874|
> > > |**+CoRA**|**0.688**|**0.791**|**0.876**|
> > > |**Moment**|0.675|0.786|0.866|
> > > |**+CoRA**|**0.681**|**0.789**|**0.873**|
> > >
> > > The results indicate that although CoRA was not specifically designed for these tasks, its direct application still yields performance improvements. This demonstrates CoRA's effectiveness in enhancing TSFMs by capturing correlation.
> > >
> > > We have included this analysis within Appendix F.8 (Lines 1245-1267).
> > >
> > > [1] TAB: Unified benchmarking of time series anomaly detection methods. PVLDB 2025
> > >
> > > [2] Deep learning for time series classification: a review. DMKD 2019
> > >
> > > ---
> > >
> > > **Thanks for your sincere suggestions again! If you have any additional questions, we can have further discussions!** 😊

---

> ### Comment · Reviewer_TnwM · 2025-11-23
> **Rating Revision Based on Rebuttal**
>
> The author provided a thorough response to my concerns, and the supplementary materials were sufficiently convincing; therefore, I have decided to raise my score.

---

> > ### Author Response · Authors · 2025-11-23
> >
> > Dear Reviewer TnwM,
> >
> > We are thrilled that our responses have effectively addressed your questions and comments. We really appreciate your efforts during this tight review timeline and the recognition of the strengths of our paper.
> >
> > Best regards,
> >
> > Authors

---

### Official Review · Reviewer_bdjv · 2025-10-28

**Soundness:** 4
**Presentation:** 3
**Contribution:** 3
**Rating:** 8
**Confidence:** 4

**Summary:**

To address the limitation that existing time series foundation models often overlook inter-channel modeling, this paper proposes a lightweight adapter, termed CoRA. CoRA innovatively decomposes the correlation matrix into time-varying and time-invariant components. It leverages learnable polynomials to capture dynamic patterns and introduces a novel dual contrastive learning mechanism to distinguish between positive and negative correlations. Optimized via a heterogeneous-local contrastive loss, CoRA incurs minimal computational overhead during the inference stage. Empirical results demonstrate that CoRA provides an effective solution for correlation modeling in multivariate time series forecasting.

**Strengths:**

1. The paper is well-motivated, addressing the critical challenge of modeling complex inter-channel dependencies in multivariate time series.
2. The manuscript is well-written, featuring clear and consistent notation throughout.
3. The empirical evaluation demonstrates that CoRA consistently surpasses state-of-the-art baselines across a diverse set of real-world datasets.

**Weaknesses:**

1. The projection layers are not much different from the related modeling methods in existing methods (such as TSMixer [1]), a more detailed clarification is needed.
2. The set of baselines for comparison is not comprehensive, as it omits recent state-of-the-art foundation models like TimerXL [2].
3. Discuss and comparison with classical baselines (e.g., PatchTST  [3], Leddam [4], iTransformer [5], Autoformer [6], DLinear [7]) is suggested.

*[1] TSMixer: An All-MLP Architecture for Time Series Forecasting.*

*[2] Timer-XL: Long-Context Transformers for Unified Time Series Forecasting.*

*[3] A Time Series is Worth 64 Words: Long-term Forecasting with Transformers.*

*[4] Revitalizing Multivariate Time Series Forecasting: Learnable Decomposition with Inter-Series Dependencies and Intra-Series Variations Modeling.*

*[5] iTransformer: Inverted Transformers Are Effective for Time Series Forecasting.*

*[6] Autoformer: Decomposition Transformers with Auto-Correlation for Long-Term Series Forecasting.*

*[7] Are Transformers Effective for Time Series Forecasting?.*

**Questions:**

1. How does CoRA, as a plug-and-play module, accommodate the fundamental architectural differences between encoder-only and encoder-decoder models, particularly with respect to their distinct training objectives and inference processes?
2. What is the underlying rationale for incorporating rule-based correlation relationships when the model is already designed to capture these dependencies through a learnable mechanism?

---

> ### Author Response · Authors · 2025-11-21
>
> Dear Reviewer bdjv, thank you for providing your detailed and constructive feedback.
>
> ---
> ### Response to W1:
> Thank you for your thoughtful comments.
>
> A key distinction from the Channel Mixer layers in TSMixer is that our projection layers operate on the feature dimension( $\mathbb{R}^d \to \mathbb{R}^d$ ） rather than the channel dimension($\mathbb{R}^N \to \mathbb{R}^N$). This design choice allows us to avoid the complexity $\mathcal{O}(N^2)$ associated with the number of channels. Furthermore, to enhance the representational power of these layers, we introduce an adaptive mechanism that aggregates contextual information over time to dynamically compute the projection weights among channels. The specifics of this mechanism are detailed in Equations 6-8.
>
> ---
> ### Response to W2:
> Thank you for your valuable and constructive comments.
> Following your suggestion, we have added more experiment on fine-tuning CoRA alone and have also evaluated the Timer-XL backbone.
>
> |Method|ETTh1|ETTh2|ETTm1|ETTm2|
> |:---|---|---|---|---|
> | **Timer-XL** | 0.439  | 0.351  | 0.355  | 0.258  |
> | **Timer-XL+CoRA** | **0.434** | **0.345** | **0.349** | **0.250** |
>
>
> ---
> ### Response to W3:
> We sincerely thank you for this constructive feedback.
> Our evaluation is specifically based on a 5% fine-tuning regime for the foundation model, which is distinct from the full training used for traditional end-to-end models. Consequently, directly comparing their performance may be inherently inequitable.
>
> Regarding channel relationship modeling, the majority of existing methods adopt a Channel-Independence strategy (PatchTST, Autoformer, DLinear).
>
> Regarding channel relationship modeling, the majority of existing methods adopt a Channel-Independence strategy. While models like Leddam and iTransformer treat individual channels as tokens to facilitate interaction via attention mechanisms, they exhibit significant limitations. First, by treating the entire sequence within a window as a single token, they fail to effectively model the Dynamic Correlation. Second, these methods do not explicitly distinguish between positive and negative correlations, thereby overlooking the Heterogeneous Correlation. Finally, they perform direct interactions across all channels without considering the partial Correlation, making them more susceptible to noise.
>
>
>
> ---
>
> ### Response to Q1:
> We appreciate this insightful question. CoRA operates as a non-invasive plugin, avoiding any alterations to the architecture or training objectives of TSFMs. Instead, it leverages the robust representations generated by the backbone, incorporates multiple correlation relationships to inject additional channel information, and performs independent predictions with these enhanced representations to augment the backbone's performance.
>
> ---
>
> ### Response to Q2:
>
> We are grateful for this opportunity to clarify the design rationale of our model.
>
> The rationale lies in the complementary nature of rule-based and learnable mechanisms. While learnable components offer flexibility, they can be sensitive to data noise and distribution shifts. The rule-based correlation matrix serves as a stable prior (or inductive bias), which helps constrain the search space and prevents the model from overfitting to transient patterns. This hybrid design significantly enhances the robustness of the correlation estimation module against distribution shifts, thereby making it more reliable for real-world TSFM deployment scenarios where non-stationarity is prevalent.
>
> ---
>
> **Thanks for your sincere suggestions again! If you have any additional questions, we can have further discussions!** 😊

---

> ### Comment · Reviewer_bdjv · 2025-11-23
> **All Clear on My Side**
>
> Thank the authors for all the revisions and clarifications. They have effectively addressed all of my concerns.
>
> I will keep my current assessment. Nice work!

---

> > ### Author Response · Authors · 2025-11-23
> >
> > Dear Reviewer bdjv:
> >
> > Thank you so much for your encouraging feedback and for your support toward the acceptance of our paper. We sincerely appreciate your time and constructive comments throughout the review process.
> >
> > Best regards,
> >
> > Authors

---

### Official Review · Reviewer_iXBW · 2025-10-30

**Soundness:** 2
**Presentation:** 2
**Contribution:** 2
**Rating:** 2
**Confidence:** 4

**Summary:**

This paper proposes a way to fine-tune multivariate time series foundation models as to better capture correlation between variables. Key ideas are to estimate a correlation matrix between features that has a time-dependent and time-invariant decomposition, and to screen for which variables to actually focus on in terms of positive or negative correlations via a contrastive learning strategy.

**Strengths:**

- I appreciate that the paper is trying to separately handle different types of correlation (DCorr, HCorr, PCorr)
- I find the idea of using a time-varying and time-invariant decomposition to be valuable, along with the idea of finding pairs of features with strong enough positive or negative correlation
- The experimental results appear to be very impressive (although as pointed out in weaknesses, I would like to see more baselines and some additional experiments)

**Weaknesses:**

- At least in how it is stated now, it's hard for me to see why Theorem 1 should hold since equation (15) in Theorem 1 disagrees with equation (5) and the decomposition showed earlier in Figure 3.
- In stating theoretical guarantees, I would suggest also providing intuition for why the guarantee should hold (this intuition should be in the main paper and not in an appendix/supplemental material as it helps the reader understand/interpret the guarantee) and whether the proof uses any nontrivial ideas, or if it's largely just based on an existing result. Maybe I'm missing something but Theorem 2 seems to be a known result regarding polynomial approximation using a variant of the mean value theorem? If the result was not previously known, what are the closest known existing theoretical results (which can help provide the reader with point(s) of comparison as to what proof techniques/ideas are novel)? Also can you comment on the extent to which the assumption holds for real data?
- Experiments: It would be helpful including more baselines, especially ones that already account for correlation structure (even if it doesn't do so as comprehensively as the proposed approach) such as UniTS and Moirai (both of which are mentioned in the paper but not actually compared against in the experiments).
- Experiments: Right now Tables 1 and 2 are reported without error bars (such as standard deviation over experimental repeats with different random seeds). Please include error bars as to give a better sense of variability in results.
- Experiments: It would be helpful seeing for the different datasets how the amount of data available for fine-tuning affects the performance of CoRA especially since I'd imagine that if one has somewhat limited data to fine-tune with, then the correlation matrix might be hard to estimate accurately.
- Overall I find that the paper has many exposition issues and typos that altogether make it so that the paper has serious clarity problems. Here are a few issues (this list is not exhaustive):
    - (minor) line ~17: since you're abbreviating "**Cor**relation-aware **A**dapter", shouldn't the abbreviation be "CorA" and not "CoRA" (i.e., why is the "R" capitalized)?
    - line ~53 (last line of page 1 that actually is one line after line 53): the explanation here makes it clear why MLP doesn't handle DCorr but why precisely does it not model HCorr? Can an MLP not figure out positive or negative correlations? I'd imagine that the learned weights would be able to capture this sort of information?
    - line ~122: "LLMs enable" should say "LLMs to enable"
    - line ~153-154: "$L$ length" should say "length $L$"
    - line ~153-154: please clearly define what $t$ is
    - line ~161: "pluged" should say "plugged"
    - line ~185: the comma after "Figure 2" should be a period
    - lines ~205-206: please clearly state what $M$ is (the rank) and perhaps comment on how it is chosen (clearly it seems like we would want $M < N$ but is this a hyperparameter that is tuned later?)
    - Figure 3: it would be helpful to better motivate the decomposition especially since the time-varying and time-invariant information is getting mixed together in the multiplication so that they're not decoupled (also Figure 3 seems to make it seem like $M_t^{corr} = Q_t V Q_t^T$ but then this isn't actually the case according to equation (5) or equation (15) later) -- yet somehow lines ~209-210 suggests that we can estimate $Q_t$ and $V$ separately and the text doesn't sufficiently provide intuition at this point for why this separate estimation should be possible
    - Section 4.1.1: in modeling dynamic regularities, it would be helpful to better motivate why polynomials are particularly well-suited for the task vs alternative approaches (I'd imagine higher order polynomials become more susceptible to noise, for instance; for example, how does the proposed approach compare with a different strategy that amounts to just using a first-order polynomial but having multiple basis functions?)
    - line ~231-232: "i-th" should say "$i$-th"
    - line ~233-234 and also line ~236: inconsistent notation -> "$q^i$" should use a bold version of "$q$" to match equation (1); also I'd suggest stating clearly that "$\odot$" stands for Hadamard product
    - line ~235: "we define the set of $C_{i,t}$" should not use the word "set" here since what is being described is not what the word "set" means in math
    - line ~250-251: "Where" should not be capitalized"
    - line ~261: "L is the size of the input series" -> "L" should be italicized (i.e., written as the math variable $L$)
    - line ~261-262: "$\mathbf{R}\in\mathbb{R}^{N\times N}$ to denote the set of $r$" -> this is not the correct use of the word "set" in math
    - equation (5) does not agree with equation (15) and also does not seem to agree with Figure 3
    - line ~282: "Where" should not be capitalized
    - line ~282: "layerNorm" should be capitalized (i.e., written as "LayerNorm") to match equations (6) and (7)
    - line ~283: $\text{MLP}_{\alpha}=\mathbb{R}^d\rightarrow\mathbb{R}^d$ ---- the equal sign "$=$" should be replaced by a colon "$:$" (a similar math typo shows up for the other MLP defined in this same line)
    - does $\beta$ in equation (7) and in equation (14) actually mean different things?

**Questions:**

Please see weaknesses (I raised a number of concerns there).

---

> ### Author Response · Authors · 2025-11-21
>
> Dear Reviewer iXBW, thank you for providing your detailed and constructive feedback.
>
> ---
> ### Response to W1:
> Thank you very much for your valuable feedback and for pointing out this issue. We would like to offer the following clarification and update the manuscript accordingly.
>
> We revised Figure 3 (Lines 204-215) by adding the rule-based correlation part $\boldsymbol{R}$, thereby aligning it more closely with Equation 5.
>
> Equation 5 defines the correlation matrix $\boldsymbol{M}^{corr}_t$ as a composition of a rule-based component $\boldsymbol{R}$ and a learnable component $\boldsymbol{Q}_t\boldsymbol{V}\boldsymbol{Q}_t^T$. Since $\boldsymbol{R}$ is a rule-based term that remains invariant within the window, our analysis primarily focuses on the learnable component $\boldsymbol{Q}_t\boldsymbol{V}\boldsymbol{Q}_t^T$.
>
> The primary purpose of Theorem 1 is to demonstrate that our method effectively decomposes the learnable component of correlation matrix into time-varying and time-invariant components. Specifically, the decomposition scheme $\boldsymbol{Q}_t\boldsymbol{V}\boldsymbol{Q}_t^T$ is functionally equivalent to the conventional additive decomposition [1, 2] like $\boldsymbol{M}_i +  \boldsymbol{M}_v$. Here $\boldsymbol{M}_i$ and $\boldsymbol{M}_v$ denote the time-invariant matrix and time-varying matrix, respectively.
>
> Accordingly, to enhance the clarity of the manuscript, we have revised the presentation of Theorem 1 (Lines 363-377) and the corresponding content in the Appendix C.1 (Lines 819-837) to focus specifically on demonstrating $\boldsymbol{Q}_t\boldsymbol{V}\boldsymbol{Q}_t^T = \boldsymbol{M}_v +  \boldsymbol{M}_i$.
>
> Theorem 1 demonstrates that $\boldsymbol{Q}_t\boldsymbol{V}\boldsymbol{Q}_t^T$ can be decomposed into the sum of a time-varying matrix and a time-invariant matrix. Consequently, the correlation matrix $\boldsymbol{M}_t^{corr}$ Equation 5 can also be decomposed into the sum of time-varying and time-invariant.
>
> [1] Graph WaveNet for Deep Spatial-Temporal Graph Modeling. IJCAI 2019
>
> [2] Enhancenet: Plugin neural networks for enhancing correlated time series forecasting. ICDE 2021

---

> ### Author Response · Authors · 2025-11-21
>
> ### Response to W2:
> We sincerely thank you for these constructive comments. Your suggestions on contextualizing our theoretical guarantees are invaluable and have helped us significantly improve the clarity of our manuscript. We address your points below:
>
> **1. Intuition for the Guarantees:**
> - **For Theorem 1:** Time-varying correlations can often be modeled as a stable, long-term component (time-invariant part) combined with dynamic fluctuations (time-varying part). A common approach is to decompose the correlation matrix into the sum of two square matrices, representing the time-invariant and time-varying components, respectively. To reduce parameter complexity, we further propose a multiplicative decomposition: $\boldsymbol{Q}_t\boldsymbol{V}\boldsymbol{Q}_t^T$. By learning the low-rank time-varying $\boldsymbol{Q}_t^T$ and time-invariant matrix $\boldsymbol{V}$, our approach attains an expressive power equivalent to the additive approach, yet operates with greater efficiency.
>
> - **For Theorem 2:** If the correlation patterns in a time series evolve smoothly over time, then this smooth evolution can be effectively approximated by a well-behaved mathematical function, such as a polynomial. Taylor's theorem provides the formal basis for this, showing that a higher-degree polynomial can capture more complex dynamics, with the approximation error decreasing accordingly.
>
> We have revised the manuscript to incorporate these intuitions (Lines 212-217 and 234-237).
>
> **2. Purpose of the Proof for Theorem 2:**
>
> We would like to clarify that the purpose of the proof for Theorem 2 is not to use nontrivial ideas, but to provide theoretical support for our primary contribution.
>
> For the learning of Dynamic Correlation, our primary contribution lies in proposing the Time-varying and Time-invariant decomposition and designing learnable polynomials to estimate the time-varying component.
>
> As for Theorem 2, which is an extension of Taylor's theorem with the Lagrange remainder term, we do not emphasize the proof itself as a major contribution; rather, it serves to theoretically demonstrate that accurate correlation estimation can be achieved through the proposed Learnable Time-aware Polynomial.
>
>
> **3. The extent to which the assumption holds for real data:**
> We would like to clarify that the smoothness assumption for $ \mathcal{F}(\boldsymbol{q})$ is well-founded in real-world scenarios, as the underlying reasons of correlation shifts, such as the continuous evolution of traffic flow and gradual system degradation, typically induce smooth drifts in correlations rather than erratic jumps.
>
> In real-world scenarios, by selecting an appropriate degree K, we can ensure that the error incurred by using learnable polynomials to approximate correlations is practically negligible without compromising computational efficiency. We leverage Theorem 2 to theoretically justify this statement:
>
> Empirically, we find that the values of both $\boldsymbol{q}$ and the derivative term $\frac{\mathcal{F}^{(K+1)}(\boldsymbol{\xi})}{(K+1)!}$ are generally within [-1, 1]. Therefore, for K=3 or 4, the error term, which contains $\boldsymbol{q}^{(K+1)}$, becomes negligible (typically < 1e-3) relative to the magnitude of $\boldsymbol{M}_t^{corr}$. This is further validated by the results in Figure 6a (Lines 1116-1126), which show that CoRA achieves superior performance around K=3 or 4, confirming that this degree is sufficient to guarantee effective application.

---

> ### Author Response · Authors · 2025-11-21
>
> ### Response to W3:
> Thank you for this valuable suggestion. To provide a more comprehensive comparison, we have conducted additional experiments with channel-dependent TSFMs (such as Moirai and UniTS). The Mean Squared Error (MSE) results are summarized in the table below.
>
> |Method|Moirai|Moirai+CoRA|UniTS|UniTS+CoRA|TTM|TTM+CoRA|
> |:-:|:-:|:-:|:-:|:-:|:-:|:-:|
> |ETT(Avg.)|0.353$\scriptstyle\pm 0.004$|0.344$\scriptstyle\pm 0.002$|0.347$\scriptstyle\pm 0.003$|0.339$\scriptstyle\pm 0.001$|0.342$\scriptstyle\pm 0.003$|**0.329**$\boldsymbol{\scriptstyle\pm 0.002}$|
> |Weather|0.257$\scriptstyle\pm 0.003$|0.241$\scriptstyle\pm 0.003$|0.235$\scriptstyle\pm 0.002$|0.220$\scriptstyle\pm 0.002$|0.226$\scriptstyle\pm 0.003$|**0.214**$\boldsymbol{\scriptstyle\pm 0.002}$|
> |AQShunyi|0.690$\scriptstyle\pm 0.003$|**0.672**$\scriptstyle\pm 0.002$|0.717$\scriptstyle\pm 0.002$|0.685$\scriptstyle\pm 0.002$|0.701$\scriptstyle\pm 0.003$|0.678$\boldsymbol{\scriptstyle\pm 0.002}$|
> |ZafNoo|0.519$\scriptstyle\pm 0.003$|0.497$\scriptstyle\pm 0.001$|0.508$\scriptstyle\pm 0.004$|0.491$\scriptstyle\pm 0.002$|0.505$\scriptstyle\pm 0.003$|**0.483$\boldsymbol{\scriptstyle\pm 0.001}$**|
>
> As the table illustrates, existing foundation models with channel-dependent mechanisms do not always achieve superior performance on downstream tasks. Furthermore, these models yield further performance improvements when integrated with CoRA.
>
> We have also added this experiment to Appendix F.4 (Lines 1188-1201).
>
> ---
>
> ### Response to W4:
> We appreciate your valuable suggestion.
> To give a better sense of variability, we have added standard deviations and confidence intervals to Table 1 (Lines 432-454) and Table 2 (Lines 473-482) in the manuscript.
>
> ---
>
> ### Response to W5:
>
> Thank you for this insightful question. To analyze how the amount of available fine-tuning data affects performance, we conducted additional experiments using varying amounts of data.
> Specifically, we fine-tune the TTM and CALF backbones on the ETTm2 and Weather datasets, using 3%, 5%, 10%, and 20% of the available training data. The MSE results are summarized in the table below.
>
> |Dataset|ETTm2||||Weather||||
> |:-:|:-:|:-:|:-:|:-:|:-:|:-:|:-:|:-:|
> |**Data percentage**|**3%**|**5%**|**10%**|**20%**|**3%**|**5%**|**10%**|**20%**|
> |**TTM**|0.263$\scriptstyle\pm 0.005$|0.259$\scriptstyle\pm 0.003$|0.256$\scriptstyle\pm 0.003$|0.250$\scriptstyle\pm 0.002$|0.237$\scriptstyle\pm 0.003$|0.226$\boldsymbol{\scriptstyle\pm 0.002}$|0.224$\scriptstyle\pm 0.002$|0.216$\boldsymbol{\scriptstyle\pm 0.001}$|
> |**TTM+CoRA**|**0.261$\scriptstyle\pm 0.004$**|**0.249$\scriptstyle\pm 0.002$**|**0.248$\scriptstyle\pm 0.001$**|**0.245$\scriptstyle\pm 0.001$**|**0.234$\scriptstyle\pm 0.003$**|**0.214$\boldsymbol{\scriptstyle\pm 0.002}$**|**0.212$\boldsymbol{\scriptstyle\pm 0.001}$**|**0.210$\boldsymbol{\scriptstyle\pm 0.001}$**|```
> |**CALF**|0.285$\scriptstyle\pm 0.005$|0.274$\scriptstyle\pm 0.003$|0.268$\scriptstyle\pm 0.004$|0.261$\scriptstyle\pm 0.003$|0.251$\scriptstyle\pm 0.004$|0.238$\boldsymbol{\scriptstyle\pm 0.002}$|0.230$\boldsymbol{\scriptstyle\pm 0.003}$|0.224$\boldsymbol{\scriptstyle\pm 0.002}$|
> |**CALF+CoRA**|**0.283$\scriptstyle\pm 0.003$**|**0.263$\scriptstyle\pm 0.002$**|**0.260$\scriptstyle\pm 0.002$**|**0.254$\scriptstyle\pm 0.002$**|**0.248$\scriptstyle\pm 0.003$**|**0.229$\boldsymbol{\scriptstyle\pm 0.002}$**|**0.223$\boldsymbol{\scriptstyle\pm 0.002}$**|**0.219$\boldsymbol{\scriptstyle\pm 0.002}$**|
>
> As the results indicate, CoRA still yields a performance improvement even in a low-data regime using only 3% of the data. Moreover, under the 5% or 10% settings, our method delivers more pronounced performance gains for TSFMs.
>
> This experiment has been added to Table 3 in the revised manuscript (Lines 514-527).

---

> ### Author Response · Authors · 2025-11-21
>
> ### Response to W6:
> We would like to express our sincerest and most profound gratitude for your exceptionally detailed and meticulous review. The time and effort you have dedicated to identifying issues related to exposition, typos, and potential logical inconsistencies are truly remarkable. Your feedback has been invaluable in helping us to significantly improve the clarity and quality of our manuscript.
>
> We have thoroughly revised and proofread the entire manuscript based on your comments. We have carefully rectified all the errors highlighted in your review, including:
>
> - (Line ~17): Corrected "**Cor**relation-aware **A**dapter" to "**CoR**relation-aware **A**dapter" to be consistent with "CoRA".
>
> - (Line ~125): Revised "LLMs enable" to "LLMs to enable."
>
> - (Line ~155 & ~271): Formatted plain text letters like "L" into their proper mathematical variable form, L.
>
> Furthermore, your detailed feedback prompted us to conduct another thorough proofread of the entire manuscript, where we identified and corrected additional similar errors. For example:
>
> - (Line ~76): Corrected "possess" to "possesses".
> - (Line ~227): Corrected "are" to "is".
> - (Line ~283): Corrected "sapces" to "spaces".
>
> We sincerely thank you again for your tremendous effort in helping us improve the quality of our manuscript.
>
> ---
>
> **Thanks for your sincere suggestions again! If you have any additional questions, we can have further discussions!** 😊

---

> ### Comment · Reviewer_iXBW · 2025-11-23
>
> Thanks for taking the time to respond to my review in detail--I really appreciate it. I have taken a look at your response, the new draft (and checked against the old draft), and also looked at the other reviews/discussion from other reviews. I think you've done a good job addressing the bulk of my concerns, and have upped my score.
>
> There are still some minor lingering issues of clarity in presentation and typos. To point out a few (I don't believe that this is exhaustive and this just comes from a quick pass over the current draft):
>
> - Right now the exposition around line 212-213 is a bit unclear. There is a sentence that says "Where, $R$ denotes the rule-based correlation matrix." Technically this is not actually a grammatically correct sentence and is a fragment and, moreover, at this point in the exposition, I don't think you've yet introduced what $R$ is at a high-level in the main text yet (perhaps when you first state it in the next, provide some sort of reference to where it will be described in detail later?). Also, the new version of Figure 3 also shows up around here and while I appreciate that $R$ has been added to the diagram, how precisely it gets combined isn't clear. How the combination happens is stated later in equation (5). Actually this makes me wonder whether the division by 2 essential in equation (5). What happens if we don't divide by 2? If you don't need to divide by 2 (or put another way, it makes no practical difference), then you would be able to simplify Figure 3 so that $R$ is combined just via simple addition. (If division by 2 is essential, you could just modify Figure 3 to say that you're taking an average to combine $R$ with $Q_t V Q_t^\top$.)
> - Lines ~372-375 should be part of Theorem 1. Please check your LaTeX so that the theorem environment ends after equation (15). After compiling, the text immediately above equation (15) should appear italicized like the rest of the theorem. A similar issue shows up in Appendix C.1.
> - Similarly, lines ~386-390 should also be part of Theorem 2 (i.e., make sure that the theorem environment ends after these lines). A similar issue shows up in Appendix C.2.
> - Line 785: "Training Phrase" -> perhaps you meant "Training Phase"
> - Line ~806: "Inference Phrase -> "Inference Phase"
> - Line ~835: "Where" should not be capitalized

---

> > ### Author Response · Authors · 2025-11-25
> >
> > Dear Reviewer iXBW:
> >
> > We sincerely appreciate the valuable suggestions you have dedicated to improving the quality of our manuscript. We are truly delighted to hear that our responses have addressed your main concerns, and we greatly appreciate your support for our work.
> >
> > To address these presentation issues and typos, we have made targeted revisions to the manuscript.
> >
> > For the exposition around line 212-213:
> >
> > - To address the **grammatical error** and **unclear phrasing** in the original sentence:
> >     'Where, $\boldsymbol{R}$ denotes the rule-based correlation matrix.' We have revised it to: 'Here, $\boldsymbol{R}\in \mathbb{R}^{N\times N}$ denotes the rule-based correlation matrix which is added to the learnable part to incorporate more prior knowledge for enhancing correlation estimation.'
> >
> > - Regarding **the division by 2** in Equation (5), our original intent was to numerically align the values with the $[−1,1]$ range of the correlation matrix $\boldsymbol{M}_t^{corr}$. However, we agree that this step is procedurally unnecessary. Therefore, we have removed this operation from Equation (5) and revised Figure 3 by adding a depiction of **additive aggregation** to more clearly illustrate the aggregation process.
> >
> > We have also resolved the issues regarding the non-standard **theorem environment** and corrected the **typos** you mentioned.
> >
> > Once again, we express our **sincere appreciation** for your feedback and your thoroughness and attention to detail. We will perform a **thorough proofreading** of the entire paper to ensure high quality.
> >
> > Best regards,
> >
> > Authors

---

### Official Review · Reviewer_k2Aw · 2025-10-31

**Soundness:** 4
**Presentation:** 4
**Contribution:** 4
**Rating:** 8
**Confidence:** 5

**Summary:**

This paper proposes CoRA (Correlation-aware Adapter), a lightweight, plug-and-play module designed to enhance existing Time Series Foundation Models (TSFMs) for multivariate forecasting. The central premise is that many TSFMs, while powerful in capturing temporal dependencies, neglect the complex inter-channel correlations that are vital for accurate multivariate prediction in downstream tasks. The main contributions of the paper can be summarized as follows:
1. A Unified Framework for Complex Correlations: The paper provides a structured view by categorizing inter-channel relationships into three types: Dynamic (DCorr), Heterogeneous (HCorr), and Partial (PCorr) correlations.
2. An Efficient and Effective Adapter Design: CoRA is designed as an efficient adapter that only requires fine-tuning. It models dynamic correlations (DCorr) by innovatively decomposing the correlation matrix into low-rank time-varying and time-invariant components.
3. A Novel Contrastive Learning Method (HPCL): The core technical contribution is the HPCL module, which models HCorr by projecting representations into separate positive and negative latent spaces. Within each space, it then applies a guided contrastive loss to cluster strongly correlated channels, a process that elegantly captures PCorr.

**Strengths:**

This paper proposes CoRA, a novel correlation-aware adapter designed to enhance the multivariate forecasting capabilities of Time Series Foundation Models (TSFMs) on specific downstream tasks.
1. Originality: The paper is the first to propose a unified framework that simultaneously addresses the dynamic, heterogeneous, and partial aspects of inter-channel correlations.
2. Quality: The paper is of high quality, featuring a rigorous methodology, clear theoretical derivations, and practical efficiency.
3. Clarity: The paper is well-written, with a complete logical flow and effective visualizations.
4. Significance: This work is highly significant for the time series domain, particularly for fine-tuning TSFMs on downstream tasks.

**Weaknesses:**

This paper could be improved in the following areas:
1. Lack of Direct Experimental Validation for Partial Correlation (PCorr): The paper claims to model three types of correlations: DCorr, HCorr, and PCorr. While experiments provide strong evidence for DCorr (e.g., Fig. 7) and HCorr (separation of positive/negative spaces), the validation for PCorr is less direct. The ablation study (Table 2) shows the overall benefit of the HPCL module but does not disentangle the specific contributions of modeling HCorr versus PCorr. The visualization in Fig. 7 implicitly shows PCorr through clustering but fails to highlight or analyze it separately.
2. Omission of Experimental Details: The paper states that the thresholdεis a "learnable" parameter but does not detail how it is optimized. Furthermore, the main results in Table 1 are based solely on a 5% few-shot setting. It is suggested that the authors include experiments conducted with full-dataset fine-tuning.
3. Confusing Notation: The authors use the standard and calligraphic forms of the same letter to represent two different concepts. It is recommended that the authors use distinct symbols and consider adding a notation table to the paper to ensure consistency and clarity.

**Questions:**

1. Regarding the Explicit Validation of Partial Correlation (PCorr): The paper's core thesis is the unified modeling of DCorr, HCorr, and PCorr. While the evidence for DCorr and HCorr is quite direct, the support for PCorr modeling appears more implicit. Could you provide a more direct piece of evidence to demonstrate that CoRA is effectively learning and leveraging partial correlations?
2. Regarding the Optimization of the Learnable Threshold ε: Could you please clarify the specific mechanism that allows gradients to flow back toεduring the training process?
3. Regarding Performance in Data-Rich Scenarios: The main experiments are conducted in a 5% few-shot setting. However, how does CoRA perform when fine-tuned on the full training dataset (100% data)? Does the performance gap between the baseline TSFM and TSFM+CoRA widen, narrow, or remain the same?

---

> ### Author Response · Authors · 2025-11-21
>
> Dear Reviewer k2Aw, thank you for providing your detailed and constructive feedback.
>
> ---
> ### Response to W1&Q1:
> Thank you very much for your valuable feedback and for pointing out this issue.
>
> The key to partial correlation is that not every pair of channels is related. Instead, each channel shows significant correlation only with a specific subset of other channels and remains independent of the rest.
>
> As shown in Figure 7 (Lines 1161-1182), the similarity we visualized is sparse in most cases. This indicates that associations are not established between all channels, which aligns with the characteristics of partial correlation.
>
> ---
> ### Response to W2&Q2:
> We appreciate you for raising this crucial point regarding the trainability of $\epsilon$.
>
> In the actual implementation, we employ a soft masking technique to ensure gradient backpropagation.
>
> Specifically, to obtain $\boldsymbol{M}_t^{pos}$, apply the following operations to $\boldsymbol{M}_t^{corr}$：
>
> $\boldsymbol{W_{mask}}= \text{sigmoid}(\frac{\boldsymbol{M}_t^{corr} - \epsilon }{1e-5})$
>
> $\boldsymbol{M}_t^{pos} = \boldsymbol{W_{mask}} \odot \boldsymbol{M}_t^{corr}$
>
> where $\boldsymbol{W_{mask}}$ denotes the weights after applying the soft mask. Specifically, values smaller than $\epsilon$ are suppressed towards 0, while those exceeding $\epsilon$ approach 1.
>
> Regarding the experiments using full-dataset fine-tuning, please refer to our response to Q3.
>
> ---
> ### Response to W3:
> We sincerely thank you for this helpful suggestion regarding mathematical notation.
>
> We have updated the manuscript to use distinct symbols for clarity. We have carefully checked the text to ensure that the same letter is not used to represent different concepts, regardless of the font style. The notation is now consistent throughout the paper.
>
> ---
>
> ### Response to Q3:
>
> We thank you for this valuable suggestion regarding the evaluation in data-rich scenarios.
>
> Due to the large-scale parameters and slow training speeds inherent to many TSFMs, it is computationally prohibitive to fine-tune all baseline models on the complete dataset. To analysis the evaluation in data-rich scenarios, we selected two computationally efficient TSFMs as backbones to perform the full-data fine-tuning experiments. The MSE results are summarized as follows:
>
> |**Data**|**5%**||**100%**||
> |:-:|:-:|:-:|:-:|:-:|
> |**Method**|**CALF**|**CALF+CoRA**|**CALF**|**CALF+CoRA**|
> |**ETT(Avg.)**|0.367|**0.356**|0.340|**0.335**|
> |**Weather**|0.238|**0.229**|0.227|**0.221**|
> |**AQShunyi**|0.732|**0.714**|0.689|**0.675**|
> |**Method**|**TTM**|**TTM+CoRA**|**TTM**|**TTM+CoRA**|
> |**ETT(Avg.)**|0.342|**0.329**|0.334|**0.327**|
> |**Weather**|0.226|**0.214**|0.220|**0.210**|
> |**AQShunyi**|0.701|**0.678**|0.682|**0.669**|
>
> Although the performance gap over the baseline narrows in data-rich scenarios, our method still achieve consistent improvements for TSFMs.
>
>
> ---
>
> **Thanks for your sincere suggestions again! If you have any additional questions, we can have further discussions!** 😊

---

### Author Response · Authors · 2025-12-02
**Rebuttal Summary (Pre-Incident Scores: 8, 6, 8, 8; Average 7.5)**

**Dear Reviewers, ACs, SACs, and PCs,**

We sincerely appreciate your dedication to this conference.

We were sorry to learn about the recent technical issues with OpenReview, and we fully support the remedial actions proposed by the committee. Fortunately, thanks to the diligence and responsiveness of our reviewers, we had essentially concluded our meaningful discussions by **Nov. 25**, well prior to the incident on **Nov. 27**.


To facilitate your final assessment, we have summarized **the score evolution**, **the acknowledged on key strengths**, and **the outcomes of the rebuttal discussion** below:

**1. Score improvement as of Nov. 25.**

|Reviewer| Changes of Score| Date of Change|Score Changes History Link|
|-|:-:|:-:|:-:|
|**k2Aw**|8 $\textcolor{green}{➜}$ 8 |-|-|
|**iXBW**|2 $\textcolor{red}{➜}$ 6|**Nov. 24**|https://openreview.net/revisions?id=hhItBSo2Mo|
|**bdjv**|8 $\textcolor{green}{➜}$ 8|-|-|
|**TnwM**|6 $\textcolor{red}{➜}$ 8|**Nov. 23**|https://openreview.net/revisions?id=et89TmFFlA|
|**Average**|6.0 $\textcolor{red}{➜}$ 7.5|**Nov. 24**|

**2. Acknowledged on Strengths.**

It is encouraging to see that reviewers agree on the following strengths of our work:

- **High Quality & Clear Motivation**: The work is well-motivated, addressing critical challenges with rigorous methodology, clear notation, and high writing quality. (bdjv, TnwM, k2Aw)

- **Novel Unified Framework**: The first to simultaneously address dynamic, heterogeneous, and partial correlations through a theoretically grounded decomposition. (k2Aw, iXBW)

- **Practical & Lightweight Design**: An efficient, plug-in design that allows easy integration with TSFMs for fine-tuning without high overhead. (k2Aw, TnwM)


**3. Summary of Discussion and Rebuttal Outcomes.**

During the rebuttal phase, we addressed the reviewers' concerns through:
- **Additional experiments**: We extended backbones and evaluated robustness in OOD, non-stationary, and high-noise settings to ensure a comprehensive evaluation.
- **Detailed theoretical explanations**: We provided intuition for **Theorems 1 & 2** and discussed the validity of our assumptions on real data.
- **Manuscript refinement**: Corrected typographical errors and improved clarity.

We are pleased that **Reviewers iXBW, bdjv, and TnwM** have explicitly confirmed that **their concerns were addressed**, supporting our work by **raising or maintaining their positive scores**.

We once again thank all reviewers for keeping active communication with us throughout the rebuttal period and for their support of our work, and we look forward to the further assessment of our submission.

Best regards,

Authors of CoRA

---

### Meta-Review · Area_Chair_75fV · 2026-01-02

**Summary:**

The submission proposes CoRA, a lightweight plug-and-play correlation-aware adapter for multivariate time series forecasting that enhances TSFMs by explicitly modeling dynamic, heterogeneous, and partial inter-channel correlations via a low-rank time-varying/time-invariant decomposition and a dual contrastive objective.

Reviewers found the paper well-motivated and clearly written, with a unified framing of correlation types and an efficient adapter design that improves forecasting performance across diverse datasets. In the rebuttal, the authors addressed the main concerns constructively by (i) adding additional experiments (including stronger baselines and robustness analyses), (ii) providing clearer theoretical intuition for the stated results and discussing assumptions, and (iii) making substantial presentation/typo fixes.

Overall, the work is technically sound, practically relevant for TSFM adaptation, and the rebuttal effectively strengthened both empirical support and clarity. I recommend accepting this paper.

**Reviewer Concerns:**

See summary.

**Reviewer Scores:**

Most of the reviewers holds the positive attitude of this paper. I think during the rebuttal & discussion, the overall score would be positive.

---

### Decision · Program_Chairs · 2026-01-26

Accept (Poster)